# EFFICIENT TIME SERIES FORECASTING VIA HYPER-COMPLEX MODELS AND FREQUENCY AGGREGATION

## ABSTRACT

Time-series forecasting is a long-standing challenge in statistics and machine learning, with one of the key difficulties being the ability to process sequences with long-range dependencies. A recent line of work has addressed this by applying the short-time Fourier transform (STFT), which partitions sequences into multiple subsequences and applies a Fourier transform to each separately. We propose the Frequency Information Aggregation (FIA-Net), a model that can utilize two backbone architectures: the Window-Mixing MLP (WM-MLP), which aggregates adjacent window information in the frequency domain, and the Hyper-Complex MLP (HC-MLP), which treats the set of STFT windows as hyper-complex (HC) valued vectors. and employ HC algebra to efficiently combine information from all STFT windows altogether. Furthermore, due to the nature of HC operations, the HC-MLP uses up to three times fewer parameters than the equivalent standard window aggre- gation method. We evaluate the FIA-Net on various time-series benchmarks and show that the proposed methodologies outperform existing state-of-the-art meth- ods in terms of both accuracy and efficiency. Our code is publicly available on `https://anonymous.4open.science/r/research-1803/`.

## 1 INTRODUCTION

Time series forecasting (TSF) is a long-standing challenge that plays a key role in various domains, such as energy management Rajagukguk et al. (2020), traffic prediction Chen & Chen (2019), and financial analysis Sezer et al. (2020). With the development of deep learning, myriad neural network (NN) architectures have been proposed and have gradually improved the accuracy on the TSF problem. Two key architectures that have been used for TSF are recurrent NNs (RNNs) Zhang & Man (1998); Graves (2012); Chung et al. (2014) and transformers Vaswani et al. (2018); Zhou et al. (2022a); Wu et al. (2021); Zhang & Yan (2023), each of which aims to capture long-term dependencies through a different functional feature extraction procedure. While both methods were proven useful, RNNs struggled with long-term dependencies Pascanu et al. (2013) or non-stationary data patterns. While transformer architectures may overlook important temporal information due to permutation invariance Kim et al. (2024), they require many parameters and may suffer from long runtime. Additional NN-based approaches for TSF consider graph NNs (GNNs) Wu et al. (2020) and decomposition models Oreshkin et al. (2019).

Recent advancements have demonstrated promising results in processing and extracting features from the frequency domain Yi et al. (2023a). Techniques leveraging frequency-based transformations have been applied in various contexts, ranging from computational efficiency improvements Wu et al. (2021) to seasonal-trend decomposition Zhou et al. (2022a). To better process the frequency domain data, Yi et al. (2023b) developed a complex-valued MLP, which demonstrated superior capability in capturing both temporal and cross-channel dependencies. To better handle non-stationarities in the data, Shen et al. (2024); Tu et al. (2024); Zeng et al. (2023b) substituted the standard FFT with the Short-Time Fourier Transform (STFT) Gabor (1946), which divides the sequence into separate windows and transforms each window individually into the frequency domain. While showing better suitability for nonstationary time series data, the STFT yields a set of windows, each of which represents exclusive information about the sequence. However, in practice, adjacent windows are highly correlated, albeit processed separately by current STFT-based models.

To incorporate the overlooked shared information, we propose the FIA-Net, a novel TSF model designed to handle long-term dependencies in the data by aggregating information from subsets of STFT windows. The FIA-Net has an MLP backbone that processes the STFT windows in the frequency domain. We propose two novel MLP architectures. The first is termed window-mixing MLP (WM-MLP), which mixes each STFT window with its neighboring bands. The second is the HC-MLP. The HC-MLP leverages HC algebra to efficiently combine information from *all* STFTs together. By using HC algebra, the FIA-Net is implemented with three times fewer parameters than the equivalent WM-MLP.

The main contributions of this paper are as follows:

- We construct the FIA-Net with the WM-MLP backbone. The resulting TSF model captures inter-window dependencies in the frequency domain and benefits from a forward pass complexity of $O(L \log L/p)$ operations, where $L$ is the lookback window length and $p$ is the number of STFT windows.

- We propose a novel HC-MLP backbone that expands the receptive field of the WM-MLP while requiring a fraction of the total parameters.

- To reduce the model size and complexity, we filter the STFT windows, leaving only the top-$M$ frequency components. We show that accuracy is maintained even when $M$ is significantly smaller than the total number of components.

- We provide an array of experiments that demonstrate the performance of the model and its efficiency. We show that the FIA-Net improves upon existing models' accuracy by up to 20%.

- We provide an ablation study, in which we explore the effect of operating over the complex plane and compare the performance of the two considered MLP backbones.

## 2 RELATED WORK

**Time-Series Forecasting** The first notable works on TSF utilize classical statistical linear models such as ARIMA. Box & Jenkins (1968); Box & Pierce (1970) which consider series decomposition. These were then generalized to a non-linear setting in Watson (1993). To overcome the limitations posed by the classical models, deep learning was incorporated, where initially, sequential deep learning was performed using RNN-based models. Two key RNN models are long-short term memory networks Graves (2012), which introduce a sophisticated gating mechanism, and the DeepAR model Zhang & Man (1998), which connects the RNN model with AR modeling. While RNNs have demonstrated expressive power for sequential modeling, they often suffer from low efficiency and high runtimes in both the forward and backward passes Pascanu et al. (2013). To address these limitations, two popular architectural advancements emerged: transformers and GNNs. Transformer-based approaches such as Informer Zhou et al. (2023), Reformer Kitaev et al. (2020), and PatchTST Nie et al. (2023) leverage the attention mechanism to effectively capture temporal dependencies while introducing innovative methods to reduce the complexity of attention operations.

In contrast, GNNs have been applied to better model dependencies among time series variables by representing them as nodes in a graph. This approach is particularly effective for capturing spatio-temporal patterns. For instance, AGCRN Bai et al. (2020) proposed an adaptive graph convolution mechanism that dynamically adjusts graph structures based on inter-series relationships. Similarly, MTGNN Wu et al. (2020) integrates graph convolutions with temporal convolutional layers to jointly learn spatial and temporal dependencies. However, GNNs were not specifically developed to improve upon RNNs but rather to address unique challenges in spatio-temporal modeling.

**Frequency Domain Models for Time Series Forecasting** A recent line of work attempts to solve the TFS problem in the frequency domain Yi et al. (2023a), with the purpose of revealing patterns that may be hidden in the time domain. The FEDformer Zhou et al. (2022a) uses a Fourier-based framework to separate trend and seasonal components by leveraging the Fourier Transform on subsequences, allowing it to isolate periodic patterns more effectively. ETSformer Woo et al. (2022) combines exponential smoothing and applies attention in the frequency domain to enhance seasonality modeling by capturing both short- and long-term dependencies. In FiLM Zhou et al. (2022b), Fourier projections are used to reduce noise and emphasize relevant features. Additionally, SFM

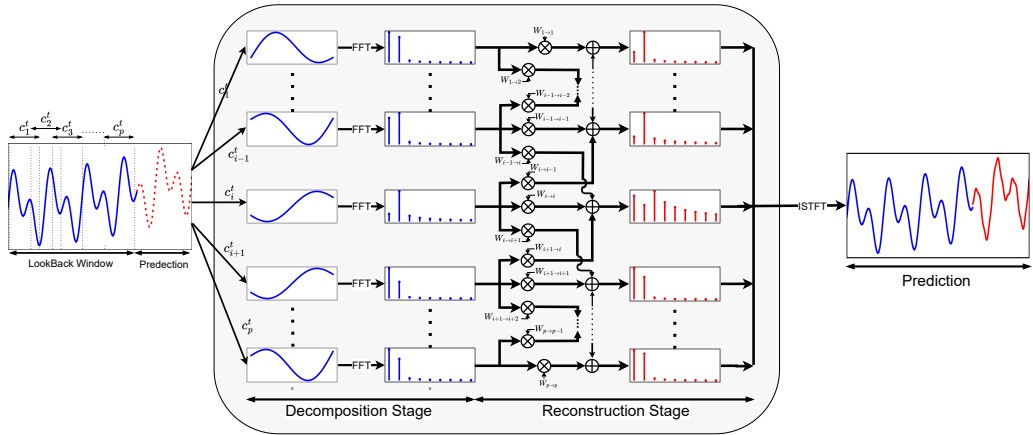

Figure 1: Window Mixing mechanism. An input $X$ is transformed into a set of $p$ STFT windows, which are transformed to the frequency domain and then fed into the WM-MLP, which aggregates adjacent windows. The WM-MLP outputs are then transformed back to the time domain via a real STFT, from which the prediction (red) is obtained.

Zhang et al. (2017) and StemGNN Cao et al. (2020) utilize frequency decomposition and Graph Fourier Transforms to handle complex temporal dependencies in multivariate time series. FRETS Yi et al. (2023b) extends this approach by proposing frequency-domain MLPs to learn complex relationships between the real and imaginary components of the FFT. FREQTSF Shen et al. (2024) uses STFT with attention mechanisms to capture temporal patterns across overlapping time windows. While frequency models, and specifically the recent use of STFT, have shown significant improvements in TFS performance, each STFT window is often processed separately, ignoring the strong correlations between adjacent windows.

**Hyper-complex Numbers** HC numbers extend the complex number system to higher dimensions Hamilton (1844). Base-4 HC numbers have been widely used in computer graphics to model $3D$ rotations Parcollet et al. (2016). Base-8 HC numbers have been explored in image classification and compression Parcollet et al. (2016); Luo et al. (2010), developing an HC network that showed favorable performance on popular datasets. The merit of HC numbers to extract relevant information in time-series was explored in Saoud & Al-Marzouqi (2020), in which an HC-net was used to analyze brainwave data, and in Kycia & Niemczynowicz (2024), which explored HC-network for financial data. In this work, we explore the utility of HC architectures for the efficient processing of STFT windows in the frequency domain.

## 3 PROPOSED MODEL : FIA-NET

In this section, we describe FIA-Net, a TSF model that leverages shared information between STFT windows. We begin by discussing the existing gap in current frequency domain TSF methods, followed by a brief introduction to frequency domain MLPs Yi et al. (2022). We then outline the FIA-Net components, presenting the novel complex MLP backbone, discussing a simple frequency compression step that reduces the MLP input dimension, and outline the complete model.

**Motivation** Even though most real-world time-series data is non-stationary, it may adhere to a piecewise stationary structure, as observed in speech signals ? and financial data Fryzlewicz & Cho (2014). This local stationarity allows us to partition the series into stationary correlated STFT subsequences that can be transformed in the frequency domain. The correlation between the STFT sequences has been efficiently utilized in recent works, even though, as we later show, it affects the downstream model accuracy in the task of time prediction.

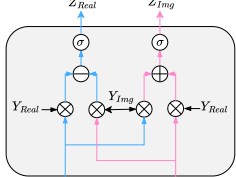

Figure 2: FD-MLP architecture.

**Frequency Domain MLPs**
As we handle complex-valued data, we adopt the frequency domain MLP (FD-MLP) unit from Yi et al. (2023b). The FD-MLP generalizes the simple neuron to operate with complex-valued weights and biases. Incorporating complex MLPs has been shown to improve the model performance as it aligns better with the geometrical structure induced by the complex plane. The FD-MLP unit is visualized in Figure 2. In Section 4, we will discuss the expansion of the FD-MLP for hyper-complex numbers.

### 3.1 ADJACENT INFORMATION AGGREGATION

Consider a sequence $X = \{x_1, \ldots, x_L\} \in \mathbb{R}^{D \times L}$ where $x_i \in \mathbb{R}^D$, $L$ is the sequence length, which we refer to as the lookback size, and $D$ is the latent space dimension. Our objective is to predict the next $T$ elements of the sequence $\hat{X} = \{\hat{x}_{L+1}, \ldots, \hat{x}_{L+T}\} \in \mathbb{R}^{D \times T}$, where $T$ is a predetermined prediction horizon. We are interested in processing $X$ in the frequency domain. We utilize the STFT, which partitions $X$ into $p$ windows and applies the FFT separately to each window. In addition, we exploit the real-valued inputs to perform a Real STFT, which results in half the frequency coefficients. The STFT for the $i$-th window is defined as:

$$\mathsf{STFT}\{X\}(\omega, \tau_i) = \sum_{t=1}^{L} x_t w(t - \tau_i) e^{-j\omega t}, \tag{1}$$

Where, $w(t - \tau_i)$ is the window function centered at the location of the $i$-th window ($i \in \{1, \ldots, p\}$), $\omega$ represents the angular frequency, and $j$ satisfies $j^2 = -1$. Each window is defined by its center $\tau_i$ and has a size of $\frac{N_{\mathsf{FFT}}}{2} + 1$. The output of the STFT consists of $p$ windows, each producing a spectrum of length $\frac{N_{\mathsf{FFT}}}{2} + 1$.

We propose the window mixing MLP (WM-MLP), which adapts the FD-MLP to properly aggregate neighboring STFT windows to incorporate shared information. Given a set of complex transformed windows $\{C_1, \ldots, C_p\}$, the WM-MLP operates on the $i$th window $C_i^{\mathsf{in}}$ as follows:

$$C_i^{\mathsf{out}} = \sigma\left(C_i^{\mathsf{in}} W_{i \to i} + C_{i-1}^{\mathsf{in}} \overline{W}_{(i-1) \to i} + C_{i+1}^{\mathsf{in}} \overline{W}_{(i+1) \to i} + B_i\right) \tag{2}$$

where $\sigma(\cdot)$ is an activation function, $(W_{(i-1) \to i}, W_{i \to i}, W_{(i+1) \to i})_{i=1}^{p}$ are the WM-MLP weight matrices with $C_j$ being a matrix of zeros for $j \notin \{1, \ldots, p\}$, and $(B_i)_{i=1}^{p}$ are the WM-MLP bias vectors, and $\overline{W}$ is the elementwise complex conjugate of $W$. The outputs of the WM-MLP are transformed back to the time domain using the element-wise inverse STFT, which is given by:

$$\mathsf{iSTFT}\{X^F(w, \tau_i)\}(t) = \sum_{\omega} X^F(\omega, \tau) e^{j\omega t} w(t - \tau_i) \tag{3}$$

The STFT, WM-MLP operation, and inverse transform are depicted by Figure 1. In highly nonstationary data, energy transition between adjacent windows can be sharp. To that end, we introduce a minor overlap between adjacent windows of $N_{FFT} - \frac{L - N_{FFT}}{p-1}$, which implicitly adjusts their statistics prior to processing by the TSF model by increasing the inter-window correlations.

### 3.2 IMPLEMENTATION DETAILS AND COMPLETE SYSTEM

**Selective Frequency Compression** To reduce the input dimensionality to the WM-MLP, we compress each transformed window $C_i \in \mathbb{C}^{N_{\mathsf{FFT}} \times D}$ along the frequency axis. Specifically, we select the top $M$ frequency components based on their real and imaginary values across each dimension and denote the compressed window with $C_i^M$. Then, $(C_1^M, \ldots, C_p^M)$ is fed into the WM-MLP layer. The top-$M$ procedure is given by

$$C_i^M = \mathop{\mathsf{Top\text{-}M}}_{j=1,\ldots,M} |C_{i,j}|_{\mathbb{C}} \tag{4}$$

where $C_{i,j}$ is the $j$th component of $C_i$ and $|z|_{\mathbb{C}}$ is the magnitude of $z \in \mathbb{C}$. Additionally, we store the top component indices of equation 4 in a list $\mathcal{I}(i)$, which encodes the band from which the information came. To transform the WM-MLP output $C_i^{\mathsf{out}}$ back to the time domain, we perform a

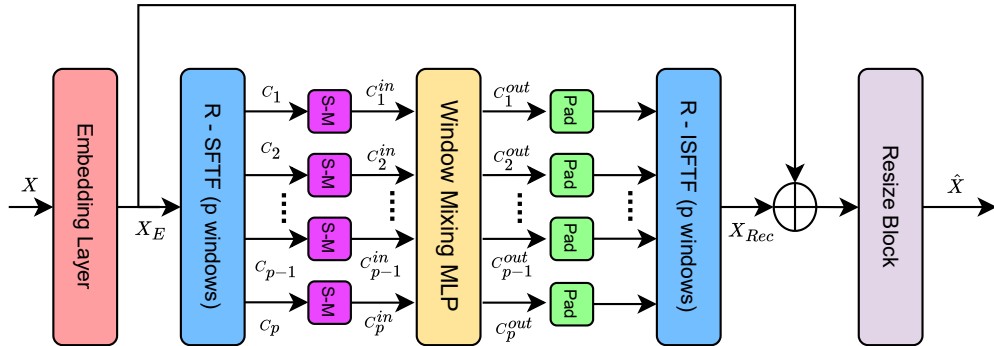

Figure 3: FIA-Net Model: The input, denoted $X$, is first fed into the embedding layer, resulting in $X_E$, which is transformed to the frequency domain via the STFT. We then extract the top-$M$ components of each STFT window and feed the compressed windows through the WM-MLP. The MLP outputs are then passed through position-aware zero padding, whose outputs are transformed back to the time domain and summed with $X_E$ via skip connection. The model output $\hat{X}$ is then given by applying a linear transformation.

*position-aware* zero padding, which adds $N_{\mathsf{FFT}} - M$ zeros while placing the nonzero components in their original indices, which correspond to the original frequency bands, i.e.,

$$C_{i,j}^{\text{padded}} = \begin{cases} C_{i,j}^{\text{out}}, & j \in \mathcal{I}(i) \\ 0, & \text{else.} \end{cases}$$

In Section 5, we demonstrate that, in addition to improving computational efficiency, this frequency compression procedure enhances the performance of downstream TSF tasks. The selection of top-$M$ components allows us to reduce the model's complexity while maintaining the most relevant frequency information.

**Complete Model** The complete FIA-Net, as shown in Figure 3, operates as follows: Given an input $X \in \mathbb{R}^{B \times L \times D}$, the dimension of $X$ is expanded through a learned embedding layer, resulting in $X_E \in \mathbb{R}^{B \times L \times D \times E}$. This expanded representation is then fed into an STFT block that uses the real input to perform $\mathsf{R} - \mathsf{STFT}$ on $X_E$. The transformed signal is passed through the SM block, whose output is further processed by the WM-MLP. The WM-MLP outputs are subsequently padded and transformed back to the temporal axis, where they are integrated with $X_E$ via a skip connection and resized to the desired output sequence shape using a two-layer MLP decomposition.

**Model Complexity** The forward pass complexity of the WM-MLP is primarily determined by the STFT complexity, which is $O(L \log(\frac{L}{p}))$. This represents a significant reduction in complexity compared to transformer-based methods, which employ intricate mechanisms to reduce their $O(L^2)$ attention complexity to $O(L \log L)$. Additionally, the application of top-$M$ frequency selection further optimizes the forward pass in the frequency domain, reducing both computational demands and the corresponding MLP size. A detailed analysis of these complexities is provided in Table 11.

## 4 WINDOW AGGREGATION VIA HYPER-COMPLEX MODELS

Even though the WM-MLP backbone integrates valuable information that benefits the FIA-Net's accuracy, information is not only shared between two adjacent STFT windows. In fact, the stronger the dependencies on the long-term past, the more information is shared between two distant windows on the frequency axis. Ideally, we would like to aggregate information between all $p$ STFT windows. Unfortunately, a straightforward extension of the WM-MLP requires $O(p^2)$ weight matrices, which may impair the training procedure and increase model complexity. To address that, we interpret the set of windows as an HC vector and propose an HC-based MLP that efficiently processes the set of STFT windows. We begin with a short introduction on HC-algebras, followed by the construction of the proposed MLP backbone for the FIA-Net.

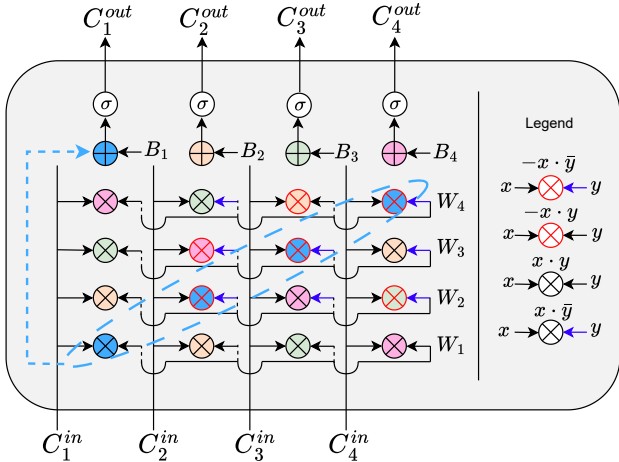

Figure 4: HC-MLP operating on $C^{\text{in}} = (C_1^{\text{in}}, C_2^{\text{in}}, C_3^{\text{in}}, C_4^{\text{in}})$, implementing the HC multiplication (equation 6). Each output unit is the sum of the corresponding inner blocks of the same color, where a $\bigoplus$ symbol denotes complex addition and a $\bigotimes$ denotes complex multiplication. A red outline denotes minus multiplication, and a blue input arrow denotes complex conjugation.

### 4.1 Hyper-complex Numbers

HC numbers generalize the complex field by introducing additional dimensions while maintaining algebraic properties. HC number systems are defined by a parameter $q$ that determines the number of components in the number system. Complex numbers can thus be viewed as an HC number with $q = 2$, and an HC number of base $q$ can be represented with $p = q/2$ complex numbers. In what follows, we focus on HC numbers with $p = 4$, termed Octonions $\mathbb{O}$, whose elements are denoted $o = (\alpha_1, \alpha_2, \alpha_3, \alpha_4) \in \mathbb{O}$, with $\alpha_i \in \mathbb{C}$ for $i = 1, \ldots, 4$. Additional discussion on $p \neq 4$ is given in Appendix C.

The addition of two Octonions, $o_1 = (\alpha_1, \ldots, \alpha_4)$ and $o_2 = (\beta_1, \ldots, \beta_4)$, is given by their componentwise sum, while their multiplication follows the Cayley-Dickson construction Khmelnytskaya & Shapiro (2021). The product $o_3 = o_1 \cdot o_2 = (\gamma_1, \gamma_2, \gamma_3, \gamma_4)$ is given by:

$$\begin{aligned}
\gamma_1 &= \alpha_1\beta_1 - \alpha_2\overline{\beta_2} - \alpha_3\overline{\beta_3} - \alpha_4\overline{\beta_4} \\
\gamma_2 &= \alpha_2\overline{\beta}_1 + \alpha_1\beta_2 + \alpha_3\overline{\beta_4} - \alpha_4\overline{\beta_3} \\
\gamma_3 &= \alpha_3\overline{\beta}_1 + \alpha_4\overline{\beta}_2 + \alpha_1\beta_3 - \alpha_2\overline{\beta}_4 \\
\gamma_4 &= \alpha_4\overline{\beta}_1 + \alpha_2\overline{\beta}_3 + \alpha_1\beta_4 - \alpha_3\overline{\beta}_2
\end{aligned} \tag{5}$$

Hyper-complex numbers exhibit additional properties such as closed-form expressions for norm calculations and norm preservation for specific bases. For completeness, we provide additional information on HC-numbers in Appendix C, where the proposed MLP is presented under specific bases.

### 4.2 Hyper-Complex MLP

The longer the range of temporal dependencies in the data, the more shared information there is between gathered windows. In such cases, the WM-MLP, which incorporates short-term information in the frequency domain, might fail to capture long-term dependencies. To that end, our goal is to increase the extent to which information is shared across the STFT windows. To derive a parameter-efficient solution, we incorporate HC algebra into the frequency domain learning procedure.

Assume that we are given $p = 4$ complex-valued STFT windows $(C_i^{\text{in}} \in \mathbb{C}^{B \times M \times E})_{i=1}^4$, where the second axis is the transformed frequency domain after top-$M$ frequency component selection. We treat the set of windows as a single Octonion tensor $(C_1^{\text{in}}, C_2^{\text{in}}, C_3^{\text{in}}, C_4^{\text{in}}) \in \mathbb{O}^{B \times M \times E}$ and feed it through an HC-valued MLP, whose output is $C^{\text{out}} = \sigma(C^{\text{in}} \cdot W + B)$. For $C^{\text{out}} =$

$(C_1^{\text{out}}, C_2^{\text{out}}, C_3^{\text{out}}, C_4^{\text{out}})$, it is given by:

$$
\begin{aligned}
C_1^{\text{out}} &= \sigma(C_1^{\text{in}} W_1 - C_2^{\text{in}} \overline{W}_2 - C_3^{\text{in}} \overline{W}_3 - C_4^{\text{in}} \overline{W}_4 + B_1), \\
C_2^{\text{out}} &= \sigma(C_2^{\text{in}} \overline{W}_1 + C_1^{\text{in}} W_2 - C_4^{\text{in}} \overline{W}_3 + C_3^{\text{in}} \overline{W}_4 + B_2), \\
C_3^{\text{out}} &= \sigma(C_3^{\text{in}} W_1 + C_1^{\text{in}} W_3 - C_2^{\text{in}} \overline{W}_4 + C_4^{\text{in}} \overline{W}_2 + B_3), \\
C_4^{\text{out}} &= \sigma(C_4^{\text{in}} \overline{W}_1 + C_1^{\text{in}} W_4 - C_3^{\text{in}} \overline{W}_2 + C_2^{\text{in}} \overline{W}_3 + B_4).
\end{aligned}
\tag{6}
$$

where $W = (W_1, \ldots, W_4) \in \mathbb{O}^{E \times E}$, $B = (B_1, \ldots, B_4) \in \mathbb{O}^{E \times 1}$ are the HC-MLP weights and bias, respectively, and $\sigma$ is a standard activation function, e.g., ReLU. We stress that, as considered in the complex MLP from Yi et al. (2023b), the HC-MLP is implemented with real-valued operations, which allows it to plug into every existing automatic differentiation scheme over standard GPUs. The HC-MLP unit is depicted in Figure 4.

The WM-MLP demonstrates distinct advantages depending on the prediction horizon. For shorter prediction lengths, it achieves better performance by effectively leveraging all available information from adjacent and nearby windows. In contrast, for longer horizons, where only closer temporal information remains relevant, the WM-MLP's ability to aggregate adjusted windows proves to be more effective. This behavior is clearly demonstrated in Section 5.2. Moreover, the HC perspective offers a significant advantage in terms of parameter efficiency. It allows for an implementation with only $p$ weight matrices, whereas the corresponding WM-MLP would require $3p - 2$ weight matrices (and even $p^2$ weight matrices for a generalization of the WM-MLP), all while preserving performance. This reduction in parameters becomes increasingly dramatic as $p > 4$, as further detailed in Appendix C.

## 5 RESULTS AND DISCUSSION

### 5.1 EXPERIMENTAL SETTING

**Datasets** Following Zhou et al. (2022a); Yi et al. (2023b), we consider the following representative real-world datasets: **1) WTH** (Weather), **2) Exchange** (Finance), **3) Traffic**, **4) ECL** (Electricity), **5) ETTh1** (Electricity transformer temperature hourly), and **6) ETTm1** (Electricity transformer temperature minutely). The train/validation/test split is 70%, 15%, and 15%, respectively.

**Baselines** In this research, we followed the TSF SoTA baselines: **1) FedFormer** Zhou et al. (2022a), **2) Reformer** Kitaev et al. (2020), **3) FreTS** Yi et al. (2023b), **4) PatchTST** Nie et al. (2023), **5) Informer** Zhou et al. (2023), **6) Autoformer** Wu et al. (2021) and **7) LSTF-Linear** Zeng et al. (2023a).

**Experiments setup** All experiments were conducted using PyTorch Paszke et al. (2019) on a single RTX 3090, utilizing mean squared error (MSE) loss and the Adam optimizer Kingma (2014). We established an initial learning rate of $10^{-3}$ with an exponential decay scheduler. Hyperparameters were optimized individually for each dataset (see Appendix B.3 for specific details). We report performance metrics under both root mean squared error (RMSE) and mean absolute error (MAE). Additional information on the Normalization B.5, datasets B.1, and baseline models B.2 can be found in the appendix.

### 5.2 MAIN RESULTS

Table 1 compares the FIA-Net performance under both the WM-MLP and the HC-MLP backbones with the SoTA baselines. It is evident that the FIA-Net consistently outperforms the baselines on most considered values of prediction horizon $T$, with an average improvement of $5.4\%$ in MAE and $3.8\%$ in RMSE over SoTA models. We note that the performance of the HC-MLP-based network, which is implemented with significantly fewer parameters, achieves comparable results with the corresponding WM-MLP and attains the best results over several settings. We can deduce that the HC-MLP is more suitable for shorter-term prediction, while the WM-MLP backbone is more suitable for longer ranges.

The WM-MLP backbone results reported in Table 1 consider an optimization with respect to $p$, the number of windows, while the HC-MLP considers a fixed size of $p = 4$ windows. Thus, for a more

Table 1: Forecasting performance comparison across datasets and prediction horizons using RMSE and MAE. Lower values indicate better performance. Bold denotes the best results, and underlined indicates the second-best.

| | Metric | Weather 96 | 192 | 336 | 720 | Exchange 96 | 192 | 336 | 720 | Traffic 96 | 192 | 336 | 720 | Electricity 96 | 192 | 336 | 720 | ETTh1 96 | 192 | 336 | 720 | ETTm1 96 | 192 | 336 | 720 |
|---|---|---|---|---|---|---|---|---|---|---|---|---|---|---|---|---|---|---|---|---|---|---|---|---|---|
| HC-MLP (Ours) | RMSE | **0.069** | **0.079** | 0.090 | 0.098 | 0.050 | 0.062 | 0.078 | 0.112 | **0.032** | 0.034 | 0.035 | 0.036 | 0.070 | 0.068 | 0.071 | 0.077 | **0.083** | **0.085** | **0.094** | **0.101** | 0.074 | 0.082 | 0.089 | 0.096 |
| | MAE | **0.030** | **0.039** | **0.043** | 0.054 | 0.035 | 0.049 | 0.061 | 0.089 | **0.016** | 0.017 | 0.017 | 0.018 | 0.040 | 0.041 | 0.044 | 0.049 | 0.057 | 0.064 | 0.068 | 0.075 | 0.049 | 0.056 | 0.060 | 0.067 |
| WM-MLP (Ours) | RMSE | 0.071 | 0.081 | **0.089** | **0.097** | 0.048 | 0.060 | 0.076 | 0.107 | 0.033 | 0.033 | 0.034 | 0.036 | 0.067 | 0.068 | 0.070 | 0.076 | 0.084 | 0.088 | 0.097 | 0.102 | 0.076 | 0.082 | 0.089 | **0.094** |
| | MAE | 0.031 | 0.041 | 0.045 | 0.053 | 0.034 | 0.047 | 0.058 | 0.086 | 0.016 | 0.016 | 0.016 | 0.018 | 0.039 | 0.041 | 0.044 | 0.049 | 0.057 | 0.066 | 0.071 | **0.075** | 0.052 | 0.055 | 0.058 | 0.064 |
| FreTS | RMSE | 0.071 | 0.081 | 0.090 | 0.099 | 0.051 | 0.067 | 0.082 | 0.110 | 0.036 | 0.038 | 0.038 | 0.039 | **0.065** | **0.064** | 0.072 | 0.079 | 0.087 | 0.091 | 0.096 | 0.108 | 0.077 | 0.083 | **0.089** | 0.096 |
| | MAE | 0.032 | 0.040 | 0.046 | 0.055 | 0.037 | 0.050 | 0.062 | 0.088 | 0.018 | 0.020 | 0.019 | 0.020 | 0.039 | 0.040 | 0.046 | 0.052 | 0.061 | 0.065 | 0.07 | 0.082 | 0.052 | 0.057 | 0.062 | 0.069 |
| PatchTST | RMSE | 0.074 | 0.084 | 0.094 | 0.102 | 0.052 | 0.074 | 0.093 | 0.166 | **0.032** | 0.035 | 0.039 | 0.040 | 0.067 | 0.066 | 0.067 | 0.081 | 0.091 | 0.094 | 0.099 | 0.113 | 0.082 | 0.085 | 0.091 | 0.097 |
| | MAE | 0.034 | 0.042 | 0.049 | 0.056 | 0.039 | 0.055 | 0.071 | 0.132 | 0.018 | 0.020 | 0.021 | 0.021 | 0.041 | 0.042 | 0.043 | 0.055 | 0.065 | 0.069 | 0.073 | 0.087 | 0.055 | 0.059 | 0.064 | 0.070 |
| LTSF-Linear | RMSE | 0.081 | 0.089 | 0.098 | 0.106 | 0.052 | 0.069 | 0.085 | 0.116 | 0.039 | 0.042 | 0.040 | 0.041 | 0.075 | 0.070 | 0.071 | 0.080 | 0.089 | 0.094 | 0.097 | 0.108 | 0.080 | 0.087 | 0.093 | 0.099 |
| | MAE | 0.040 | 0.048 | 0.056 | 0.065 | 0.038 | 0.053 | 0.064 | 0.092 | 0.020 | 0.022 | 0.020 | 0.021 | 0.045 | 0.043 | 0.044 | 0.054 | 0.063 | 0.067 | 0.070 | 0.082 | 0.055 | 0.060 | 0.065 | 0.072 |
| FEDformer | RMSE | 0.088 | 0.092 | 0.101 | 0.109 | 0.067 | 0.082 | 0.105 | 0.183 | 0.036 | 0.042 | 0.042 | 0.042 | 0.072 | 0.072 | 0.075 | 0.077 | 0.096 | 0.100 | 0.105 | 0.116 | 0.087 | 0.093 | 0.102 | 0.108 |
| | MAE | 0.050 | 0.051 | 0.057 | 0.064 | 0.050 | 0.064 | 0.080 | 0.151 | 0.022 | 0.023 | 0.022 | 0.022 | 0.049 | 0.049 | 0.051 | 0.055 | 0.072 | 0.076 | 0.080 | 0.090 | 0.063 | 0.068 | 0.075 | 0.081 |
| Autoformer | RMSE | 0.104 | 0.103 | 0.101 | 0.110 | 0.066 | 0.083 | 0.101 | 0.181 | 0.042 | 0.050 | 0.053 | 0.050 | 0.075 | 0.099 | 0.115 | 0.119 | 0.105 | 0.114 | 0.119 | 0.136 | 0.109 | 0.112 | 0.125 | 0.126 |
| | MAE | 0.064 | 0.061 | 0.059 | 0.065 | 0.050 | 0.063 | 0.075 | 0.150 | 0.026 | 0.033 | 0.034 | 0.035 | 0.051 | 0.051 | 0.088 | 0.116 | 0.079 | 0.086 | 0.088 | 0.102 | 0.081 | 0.083 | 0.091 | 0.093 |
| Informer | RMSE | 0.139 | 0.134 | 0.115 | 0.132 | 0.084 | 0.088 | 0.127 | 0.170 | 0.039 | 0.047 | 0.053 | 0.054 | 0.124 | 0.138 | 0.144 | 0.148 | 0.121 | 0.137 | 0.145 | 0.157 | 0.096 | 0.107 | 0.119 | 0.149 |
| | MAE | 0.101 | 0.097 | 0.101 | 0.132 | 0.066 | 0.068 | 0.093 | 0.117 | 0.023 | 0.030 | 0.034 | 0.035 | 0.094 | 0.105 | 0.112 | 0.116 | 0.093 | 0.103 | 0.112 | 0.125 | 0.070 | 0.082 | 0.090 | 0.115 |
| Reformer | RMSE | 0.152 | 0.201 | 0.203 | 0.228 | 0.146 | 0.169 | 0.189 | 0.201 | 0.053 | 0.054 | 0.053 | 0.054 | 0.125 | 0.138 | 0.144 | 0.148 | 0.143 | 0.148 | 0.155 | 0.155 | 0.089 | 0.108 | 0.128 | 0.163 |
| | MAE | 0.108 | 0.147 | 0.154 | 0.173 | 0.126 | 0.147 | 0.157 | 0.166 | 0.035 | 0.035 | 0.035 | 0.035 | 0.095 | 0.121 | 0.122 | 0.120 | 0.113 | 0.120 | 0.124 | 0.126 | 0.065 | 0.081 | 0.100 | 0.132 |

suitable comparison, Table 2 shows a comparison of the FIA-Net performance under both backbones with $p = 4$. We note that when $p$ is similar for both models, the FIA-Net attains similar results under both backbones, while the HC-MLP requires significantly fewer parameters. Consequently, when the number of windows allows for an HC-MLP version (e.g., $p = 2^\ell$ as we further explain in Appendix C), an HC-MLP backbone is preferable.

Table 2: Performance comparison between WM-MLP and HC-MLP with a fixed number of STFT windows (p = 4). Results demonstrate that HC-MLP achieves comparable accuracy while significantly reducing model parameters, making it preferable for efficient implementations.

| | Metric | Traffic 96 | 192 | 336 | 720 | ETTh1 96 | 192 | 336 | 720 | ETTm1 96 | 192 | 336 | 720 |
|---|---|---|---|---|---|---|---|---|---|---|---|---|---|
| WM-MLP ($p = 4$) | RMSE | 0.033 | 0.034 | 0.035 | 0.036 | 0.088 | 0.094 | 0.100 | 0.103 | 0.074 | 0.082 | 0.089 | 0.096 |
| | MAE | 0.016 | 0.016 | 0.017 | 0.018 | 0.058 | 0.064 | 0.068 | 0.075 | 0.049 | 0.056 | 0.060 | 0.067 |
| HC-MLP | RMSE | 0.032 | 0.034 | 0.035 | 0.036 | 0.083 | 0.085 | 0.094 | 0.101 | 0.072 | 0.082 | 0.089 | 0.096 |
| | MAE | 0.016 | 0.017 | 0.017 | 0.018 | 0.049 | 0.057 | 0.064 | 0.068 | 0.049 | 0.056 | 0.060 | 0.067 |

## 5.3 ABLATION STUDIES

We consider three ablation studies that best demonstrate the key aspects of the proposed work. We focus on the effect of frequency selection, the size of the lookback window, and the omission of real/imaginary components in the training procedure. We show that, in various cases, the total amount of parameters can be decreased by up to 60%. Due to space limitations, the results are demonstrated on a single dataset, while a full discussion and additional results are given in Appendix D.4.

### 5.3.1 FREQUENCY DIMENSION COMPRESSION

We study the effect of the parameter $M$ in the top-$M$ frequency component selection process on the ETTh dataset. As seen in figure 5, even though the model performance varies over different datasets and forecasting horizon sizes, in most cases, $M = 4$ attains the best accuracy. Furthermore, note that taking $M < M_{\max} = \frac{N_{FFT}}{2} + 1$ improves the model's results. We conjecture that considering fewer frequency components decreases the NN class complexity, which potentially simplifies the optimization procedure landscape while preserving most of the information contained within the signal. We expand upon this discussion and provide additional results in the Appendix D.1.

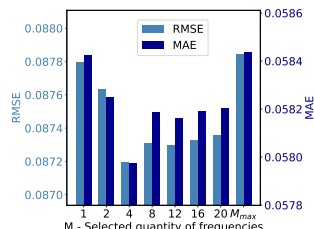

Figure 5: Accuracy vs. $M$

### 5.3.2 EFFECT OF LOOKBACK WINDOW SIZE

In this section, we evaluate the impact of varying lookback window sizes $L \in \{24, 48, 96, 192, 288, 480, 576, 720\}$ for different prediction lengths $T \in \{96, 192, 336, 720\}$. As shown in Figure 6, the dotted line represents the RMSE, while the solid line represents the MAE. The model's performance initially improves as $L$ increases, as expected, since a longer lookback provides more contextual information. However, many models exhibit parabolic behavior, where performance deteriorates after a certain point due to overfitting to noise or unrealistic patterns in the data. In contrast, our model maintains stable performance and effectively avoids overfitting, demonstrating its robustness to changes in lookback window size. Additional experiments can be found in Appendix D.2.

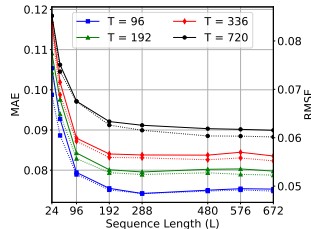

Figure 6: Accuracy vs. $L$.

### 5.3.3 REDUNDANCY OF COMPLEX REPRESENTATION

We study the effect of the real and imaginary components on prediction quality. We fix the hyperparameters $E = 128$, $p = 13$, $N_{FFT} = 16$, $M = M_{\text{max}}$, and compare several scenarios, such that each scenario considers the masking of a different component, either in the data, the parameters, or both. The masking occurs in both training and inference. As seen in Table 3, the elimination of either the real or imaginary components in the data does not significantly affect the downstream accuracy, which may hint at redundancy in the learning procedure. Furthermore, this redundancy is maintained when we consider the intersection omission of the real/imaginary parts of both the data and the MLP weights. This phenomenon can be explained through the Kramers-Kronig relation (KKR) Kronig (1926); Kramers (1927), which provides a representation of the real component of an analytic complex-valued function in terms of its complex components and vice versa. Roughly speaking, for a complex-valued function $c(\omega) = \text{Re}\{c\}(\omega) + i\text{Im}\{c\}(\omega)$, the KKR are given by

$$\text{Re}\{c\}(\omega) = \frac{1}{\pi} \int_{-\infty}^{\infty} \frac{\text{Im}\{c\}(\sigma)}{\omega - \sigma} d\sigma, \quad \text{Im}\{c\}(\omega) = -\frac{1}{\pi} \int_{-\infty}^{\infty} \frac{\text{Re}\{c\}(\sigma)}{\omega - \sigma} d\sigma.$$

Thus, we conjecture that masking one component forces the other to recover both in the learning procedure by implicitly approximating the KRR. We therefore believe that a sophisticated system design that considers a KRR-based architecture may lead to the sufficiency of a single component in the forecasting task but leaves a complete study of that subject to future work. This phenomenon is further explored in Appendix 8.

| Dataset | I/O Hidden Part | 96/96 MAE | 96/96 RMSE | 96/192 MAE | 96/192 RMSE | 96/336 MAE | 96/336 RMSE | 96/720 MAE | 96/720 RMSE |
|---|---|---|---|---|---|---|---|---|---|
| ETTm1 | $X^{\text{Real}}$ | 0.0522 | 0.0797 | 0.0560 | 0.0850 | 0.0597 | 0.0888 | 0.0658 | 0.0958 |
| | $X^{\text{Imag}}$ | 0.0521 | 0.0792 | 0.0562 | 0.0844 | 0.0592 | 0.0879 | 0.0684 | 0.0976 |
| | $W^{\text{Real}}$ | 0.0522 | 0.0791 | 0.0557 | 0.0843 | 0.0588 | 0.0875 | 0.0669 | 0.0964 |
| | $W^{\text{Imag}}$ | 0.0526 | 0.0801 | 0.0560 | 0.0849 | 0.0596 | 0.0888 | 0.0651 | 0.0953 |
| | $W^{\text{Imag}}, X^{\text{Imag}}$ | 0.0523 | 0.0798 | 0.0560 | 0.0849 | 0.0592 | 0.0884 | 0.0644 | 0.0947 |
| | $W^{\text{Real}}, X^{\text{Real}}$ | 0.0522 | 0.0791 | 0.0557 | 0.0843 | 0.0588 | 0.0887 | 0.0669 | 0.0930 |
| | $\emptyset$ | 0.0522 | 0.0791 | 0.0565 | 0.0848 | 0.0592 | 0.0878 | 0.0685 | 0.0975 |

Table 3: Performance comparison on ETTm1 for $I/O = 96 \times \{96, 192, 336, 720\}$ with various modes. $X^{\text{Real}}/X^{\text{Imag}}$ hide the real/imaginary parts of the input, while $W^{\text{Real}}/W^{\text{Imag}}$ zero out the corresponding weights. Completely ignoring both components is denoted as $(W^{\text{Imag}}, X^{\text{Imag}})$ or $(W^{\text{Real}}, X^{\text{Real}})$.

## 6 CONCLUSION

This paper presents FIA-Net, a new model for long-term time series forecasting using STFT window aggregation in the frequency domain and HC MLPs. The proposed methodology shows superior performance over existing SoTA on standard benchmark datasets. We show that treating the set of

STFT windows as a single HC tensor, which is processed by a novel HC-MLP, significantly reduces the total amount of parameters, with no degradation in the TSF accuracy. We study various schemes to increase model efficiency by, for example, choosing the top-$M$ magnitude frequency components. Experimental results show that the omission of one of the complex representation components does not induce notable segregation in performance, which may be explained by the KKR. For future work, we aim to leverage the KKR equations to propose a forecasting model that only considers the real component in the complex representation while operating over the complex plane. Additionally, we plan to further investigate the relationship between the number of adjacent STFT windows in the WM-MLP backbone and the statistical properties of the datasets.

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

# APPENDIX FOR "EFFICIENT TIME SERIES FORECASTING VIA HYPER-COMPLEX MODELS AND FREQUENCY AGGREGATION"

## A NOTATIONS & SYMBOLS

### A.1 NOTATION

We provide a detailed table of the involved notation in this paper:

| Symbol | Description |
|---|---|
| $B$ | Batch size. |
| $L$ | Lookback window size. |
| $D$ | Number of features for each time step. |
| $T$ | Length of the prediction horizon. |
| $E$ | Embedding size. |
| $M$ | Number of frequencies to select from all the frequencies using the top M magnitudes. |
| $X$ | Multivariate time series with a lookback window of size $L$ at timestamps $t$. |
| $X_t$ | Multivariate values of $D$ distinct series at timestamp $t$. |
| $X_{t,i}$ | The value of the $i$-th feature of the distinct series at timestamp $t$. |
| $\hat{X}$ | Ground truth target values. |
| $\sigma$ | activation function |
| $P$ | Number of windows in the STFT. |
| $N_{FFT}$ | Number of frequency bins in each window of the STFT. |
| $\omega$ | Window function for the STFT. |
| $X_E$ | X after traversing through the embedding layer. |
| $X_{Rec}$ | The reconstructed X after the frequency alteration. |
| $c_i^t$ | The $i$-th window of the input in the time domain. |
| $C_i$ | The $i$-th window of the STFT containing $N_{FFT}$ frequency bins. |
| $C_i^{\text{in}}$ | The $i$-th window of the STFT, retaining the top $M$ frequency components based on magnitude. |
| $C_i^{\text{out}}$ | The $i$-th window of the STFT after the WM-MLP/WHC has been applied. |
| $W_{i \to j}$ | The weights that capture the frequency energy shift between window $i$ and $j$, defined as $W_{i \to j} = W_{i \to j}^{\text{Real}} + jW_{i \to j}^{\text{Img}}$, where $W_{i \to j} \in \mathbb{C}^{E \times E}$. |
| $B_{i \to j}$ | The bais that capture the frequency energy shift between windows $i$ and $j$, defined as $B_{i \to j} = B_{i \to j}^{\text{Real}} + jB_{i \to j}^{\text{Img}}$, where $B_{i \to j} \in \mathbb{C}^{E}$. |

Table 4: Table of Symbols and Descriptions

### A.2 DIMENSIONS

The following table summarizes the dimensions of the data tensor in every step of the FIA-Net.

| Symbol | Dimension |
|---|---|
| $X$ | $\mathbb{R}^{B \times L \times D}$ |
| $X_E$ | $\mathbb{R}^{B \times L \times D \times E}$ |
| $C_i$ | $\mathbb{C}^{B \times N_{FFT} \times D \times E}$ |
| $C_i^M$ | $\mathbb{C}^{B \times M \times D \times E}$ |
| $C_i^{\text{in/out}}$ | $\mathbb{C}^{B \times M \times D \times E}$ |
| $X_{Rec}$ | $\mathbb{R}^{B \times L \times D \times E}$ |
| $\hat{X}$ | $\mathbb{R}^{B \times T \times D}$ |

Table 5: Table of Symbols and Dimension

## B  ADDITIONAL EXPERIMENTAL DETAILS

### B.1  DATASET DESCRIPTIONS

In our experiments, we utilized thirteen real-world datasets to assess the effectiveness of models for long-term TSF. Below, we provide the details of these datasets, categorized by their forecasting horizon.

- **Exchange**: This dataset includes daily exchange rates for eight countries (Australia, Britain, Canada, Switzerland, China, Japan, New Zealand, and Singapore) from 1990 to 2016.

- **Weather**: This dataset gathers 21 meteorological indicators, including humidity and air temperature, from the Weather Station of the Max Planck Biogeochemistry Institute in Germany in 2020. The data is collected every 10 minutes.

- **Traffic**: For long-term forecasting, this dataset includes hourly traffic data from 862 free-way lanes in San Francisco, with data collected since January 1, 2015.

- **Electricity**: For long-term forecasting, this dataset covers electricity consumption data from 321 clients, with records starting from January 1, 2011, and a sampling interval of 15 minutes.

- **ETT**: This dataset is sourced from two electric transformers, labeled ETTh1 and ETTm1, with two different resolutions: 15 minutes and 1 hour. These are used as benchmarks for long-term forecasting.

| Datasets | Weather | Traffic | Electricity | ETTh1 | ETTm1 | Exchange Rates |
|---|---|---|---|---|---|---|
| Features | 21 | 862 | 321 | 7 | 7 | 8 |
| Timesteps | 52696 | 17544 | 26304 | 17420 | 69680 | 7588 |
| Frequency | 10m | 1h | 1h | 1h | 15m | 1d |
| Lookback Window | 96 | 48 | 96 | 96 | 96 | 96 |
| Prediction Length | 96, 192, 336, 720 | 96, 192, 336, 720 | 96, 192, 336, 720 | 96, 192, 336, 720 | 96, 192, 336, 720 | 96, 192, 336, 720 |

Table 6: Long Term Datasets Parameters

### B.2  BASELINES

We employ a selection of SoTA representative models for our comparative analysis, focusing on Transformer-based architectures and other popular models. The models included are as follows:

- **Informer**: Informer enhances the efficiency of self-attention mechanisms to effectively capture dependencies across variables. The source code was obtained from GitHub, and we utilized the default configuration with a dropout rate of 0.05, two encoder layers, one decoder layer, a learning rate of 0.0001, and the Adam optimizer.

- **Reformer**: Reformer combines the power of Transformers with efficient memory and computation management, especially for long sequences. The source code was sourced from GitHub, and we employed the recommended configuration for our experiments.

- **Autoformer**: Autoformer introduces a decomposition block embedded within the model to progressively aggregate long-term trends from intermediate predictions. The source code was accessed from GitHub, and we followed the recommended settings for all experiments.

- **FEDformer**: FEDformer introduces an attention mechanism based on low-rank approximation in the frequency domain combined with a mixture of expert decomposition to handle distribution shifts. The source code was retrieved from GitHub. We utilized the Frequency Enhanced Block (FEB-f) and selected the random mode with 64 as the experimental configuration.

- **LTSF-Linear**: LTSF-Linear is a minimalist model employing simple one-layer linear models to learn temporal relationships in time series data. We used it as our baseline for long-term forecasting, downloading the source code from GitHub, and adhered to the default experimental settings.

- **PatchTST**: PatchTST is a Transformer-based model designed for TSF, introducing patching and a channel-independent structure to enhance model performance. The source code was obtained from GitHub, and we used the recommended settings for all experiments.

- **FreTS**: FRETS is a sophisticated model tailored for efficient TSF by exploiting a frequency domain approach. The implementation is available on GitHub, and we utilized the default configuration as recommended by the authors. In our work, FRETS serves as the foundational model. We address its limitations, particularly its handling of non-stationary data, while adapting its strengths, such as its complex frequency learner. To fully grasp the contributions of this paper, we recommend reviewing FRETS in detail first.

## B.3 IMPLEMENTATION DETAILS

Table 7 lists the hyperparameter values used in the FIA-Net implementation. Both WM-MLP and HC-MLP backbones are implemented with the same hyperparameter values, except for $p$, the number of STFT windows.

| DataSets | Weather | Traffic | Electricity | ETTh1 | ETTm1 | Exchange rate |
|---|---|---|---|---|---|---|
| Batch Size | 16 | 4 | 4 | 8 | 8 | 8 |
| Embed Size | 128 | 32 | 64 | 128 | 128 | 128 |
| Hidden Size | 256 | 256 | 256 | 256 | 256 | 256 |
| NFF | 16 | 32 | 32 | 6 | 48 | 32 |
| STFT Windows | 7 | 13 | 13 | 33 | 4 | 13 |
| S-M | 10 | $M_{\max}$ | 4 | 4 | 4 | $M_{\max}$ |
| Epoch | 10 | 10 | 10 | 10 | 10 | 10 |

Table 7: Hyperparameter Settings for Long-Term Datasets for the WM-MLP and HC-MLP

## B.4 EVALUATION METRICS

In this study, we use the Mean Squared Error (MSE) as the loss function during training. However, for evaluation, we report both the Mean Absolute Error (MAE) and the Root Mean Squared Error (RMSE).

which are defined as follows:

$$\text{MSE} = \frac{1}{n}\sum_{i=1}^{n}(Y_i - \hat{Y}_i)^2, \quad \text{RMSE} = \sqrt{\frac{1}{n}\sum_{i=1}^{n}(Y_i - \hat{Y}_i)^2}, \quad \text{MAE} = \frac{1}{n}\sum_{i=1}^{n}|Y_i - \hat{Y}_i|$$

Where:

- $Y_i$ represents the true target values,

- $\hat{Y}_i$ represents the predicted values,

- $n$ is the total number of samples.

## B.5 NORMALIZATION METHODS

In this study, similar to the FRETS model Yi et al. (2023b), we apply min-max normalization to standardize the input data to the range between 0 and 1. This method helps in ensuring that all features contribute equally to the model and prevents any specific feature from dominating due to differences in scale. The formula for min-max normalization is given by:

$$X_{Norm} = \frac{X - X_{\min}}{X_{\max} - X_{\min}}$$

By normalizing the data, we ensure that all input features are within the same range, which can improve model convergence and performance.

## C  ADDITIONAL INFORMATION ON HC NUMBERS AND MODELS

In this section we extend the discussion on HC numbers, considering additional values of $p$ beyond $p = 4$. We couple the presentation with the construction of the corresponding HC-MLP in the considered base. Recall that the base of a HC number, i.e., the number of its components is given by $b = 2p$. While hyper-complex number can be defined for any value of $b$, most research has been performed on $b$ that is given by a power of 2, as the resulting structure of the (algebraic) field. The addition of two HC numbers is simply given by the component-wise summation. In what follows, we focus on HC multiplication and additional properties. For more information on the HC number system,, we refer the reader to Kantor & Solodovnikov (1989).

### C.1  BASE 2 - COMPLEX NUMBERS

When $b = 2$, the resulting field is the complex plane $\mathbb{C}$. We describe $\mathbb{C}$ for completeness of presentation. Given two complex numbers $C_1 = \alpha_1 + j\alpha_2$ and $C_2 = \beta_1 + j\beta_2$, where $\alpha_1, \alpha_2, \beta_1, \beta_2$ are real numbers, their complex multiplication is defined as:

$$C_1 \cdot C_2 = (\alpha_1\beta_1 - \alpha_2\beta_2) + j(\alpha_1\beta_2 + \alpha_2\beta_1)$$

The norm of a complex number is given by:

$$|C_1|_{\mathbb{C}} = \sqrt{\alpha_1^2 + \alpha_2^2},$$

which is preserved under multiplication, i.e.,

$$|C_1 \cdot C_2|_{\mathbb{C}} = |C_1|_{\mathbb{C}} \cdot |C_2|_{\mathbb{C}}.$$

Since the STFT with a single window ($p = 1$) is equivalent to the standard FFT, applying our method for hyper-complex number MLP results in the following equation:

$$C_{\text{in}} = \text{FFT}(X)$$

$$C^{\text{out}} = \sigma(C_{\text{Real}}^{\text{in}} \cdot W_{1,\text{Real}} - C_{\text{Imag}}^{\text{in}} \cdot W_{1,\text{Imag}} + B_{1,\text{Real}}) + \sigma(j(C_{\text{Real}}^{\text{in}} \cdot W_{1,\text{Imag}} + C_{\text{Imag}}^{\text{in}} \cdot W_{1,\text{Real}} + B_{1,\text{Imag}})$$

Here, $W_i \in \mathbb{C}^{E \times E}$ denotes the layer weights, $B \in \mathbb{C}^E$ represents the bias term, and the multiplication occurs across the embedding dimension. Note that for $b = 2$ the HV formulation boils down to the one from Yi et al. (2023b). Thus, the HC-MLP can be considered as an HC generalization of the FD-MLP. which allows for efficient window aggregation.

### C.2  BASE 4 - QUATERNIONS

Denote the field of Quaternions with $\tilde{\mathbb{Q}}$. We represent Quatenions with a couple of Complex number, i.e., for $H_1, H_2 \in \tilde{\mathbb{Q}}$, $H_1 = (\alpha_1, \alpha_2)$ and $H_2 = (\beta_1, \beta_2)$, their multiplication is defined as

$$H_1 \cdot H_2 = (\alpha_1\beta_1 - \overline{\alpha_2}\beta_2, \quad \alpha_2\overline{\beta_1} + \alpha_1\beta_2)$$

The norm of a quaternion is given by:

$$|q|_{\tilde{\mathbb{Q}}} = \sqrt{|\alpha_1|_{\mathbb{C}}^2 + |\alpha_2|_{\mathbb{C}}^2}$$

The norm is preserved under multiplication, meaning:

$$|q_1 \cdot q_2|_{\tilde{\mathbb{Q}}} = |q_1|_{\tilde{\mathbb{Q}}} \cdot |q_2|_{\tilde{\mathbb{Q}}}$$

For our model, the corresponding HC-MLP (which we denote QuatMLP) operating on $C^{\text{in}} = (C_1^{\text{in}}, C_2^{\text{in}}) \in \tilde{\mathbb{Q}}$, is given by,

$$C^{\text{out}} = \text{QuatMLP}(C^{\text{in}}) = \sigma(C^{\text{in}} \cdot W + B)$$

where:

$$C_1^{\text{out}} = \sigma(C_1 \cdot W_1 - \overline{C_2} \cdot W_2 + B_1), \quad C_2^{\text{out}} = \sigma(C_2 \cdot \overline{W_1} + C_1 \cdot W_2 + B_2).$$

Here, $W_i \in \mathbb{C}^{E \times E}$, $i = 1, 2$ denote the layer weights, $B \in \mathbb{C}^E$ represents the bias term, and the multiplication involves complex MLP operations across the embedding dimension.

## C.3  BASE 16 - SEDENIONS

Elements on the Sedenions field, denoted $SS$, are denoted with 8-tuples of complex numbers. Given two sedenions represented by complex numbers $S_1, S_2 \in SS$, $S_1 = (\alpha_1, \alpha_2, \ldots, \alpha_8)$ and $S_2 = (\beta_1, \beta_2, \ldots, \beta_8)$, their multiplication is given by

$$
S_1 \cdot S_2 = \begin{pmatrix}
\alpha_1\beta_1 - \alpha_2\overline{\beta_2} - \alpha_3\overline{\beta_3} - \alpha_4\overline{\beta_4} - \alpha_5\overline{\beta_5} - \alpha_6\overline{\beta_6} - \alpha_7\overline{\beta_7} - \alpha_8\overline{\beta_8} \\
\alpha_1\beta_2 + \alpha_2\beta_1 + \alpha_3\overline{\beta_4} - \alpha_4\overline{\beta_3} + \alpha_5\overline{\beta_6} - \alpha_6\overline{\beta_5} + \alpha_7\overline{\beta_8} - \alpha_8\overline{\beta_7} \\
\alpha_1\beta_3 - \alpha_2\overline{\beta_4} + \alpha_3\beta_1 + \alpha_4\beta_2 + \alpha_5\overline{\beta_7} - \alpha_6\overline{\beta_8} - \alpha_7\overline{\beta_5} + \alpha_8\overline{\beta_6} \\
\alpha_1\beta_4 + \alpha_2\beta_3 - \alpha_3\beta_2 + \alpha_4\beta_1 + \alpha_5\overline{\beta_8} + \alpha_6\overline{\beta_7} - \alpha_7\overline{\beta_6} - \alpha_8\overline{\beta_5} \\
\alpha_1\beta_5 - \alpha_2\overline{\beta_6} - \alpha_3\overline{\beta_7} - \alpha_4\overline{\beta_8} + \alpha_5\beta_1 + \alpha_6\beta_2 + \alpha_7\beta_3 + \alpha_8\beta_4 \\
\alpha_1\beta_6 + \alpha_2\beta_5 - \alpha_3\overline{\beta_8} + \alpha_4\overline{\beta_7} - \alpha_5\beta_2 + \alpha_6\beta_1 - \alpha_7\beta_4 + \alpha_8\beta_3 \\
\alpha_1\beta_7 + \alpha_2\beta_8 + \alpha_3\beta_5 - \alpha_4\beta_6 - \alpha_5\beta_3 + \alpha_6\beta_4 + \alpha_7\beta_1 - \alpha_8\beta_2 \\
\alpha_1\beta_8 - \alpha_2\beta_7 + \alpha_3\beta_6 + \alpha_4\beta_5 - \alpha_5\beta_4 - \alpha_6\beta_3 + \alpha_7\beta_2 + \alpha_8\beta_1
\end{pmatrix}
$$

where each component follows the rules of Complex multiplication. The norm of a sedenion is given by:

$$
|S|_{SS} = \sqrt{\sum_{j=1}^{8} |\alpha_j|_{\mathbb{C}}^2}
$$

Unlike nase 2, 4 and 8, Sedenions *do not* preserve the norm under addition and multiplication.

The base-16 HC-MLP, denoted SedMLP, operating on an input $C^{\mathsf{in}}$ from the STFT with multiple windows $C^{\mathsf{in}} = (C_j^{\mathsf{in}})_{j=1}^{8}$, is given by

$$
C^{\mathsf{out}} = \mathrm{SedMLP}(C^{\mathsf{in}}) = \sigma(C^{\mathsf{in}} \cdot W + B)
$$

where:

$$C_1^{\mathsf{out}} = \sigma\left(C_1^{\mathsf{in}}W_1 - C_2^{\mathsf{in}}\overline{W_2} - C_3^{\mathsf{in}}\overline{W_3} - C_4^{\mathsf{in}}\overline{W_4} - C_5^{\mathsf{in}}\overline{W_5} - C_6^{\mathsf{in}}\overline{W_6} - C_7^{\mathsf{in}}\overline{W_7} - C_8^{\mathsf{in}}\overline{W_8} + B_1\right)$$

$$C_2^{\mathsf{out}} = \sigma\left(C_1^{\mathsf{in}}W_2 + C_2^{\mathsf{in}}W_1 + C_3^{\mathsf{in}}\overline{W_4} - C_4^{\mathsf{in}}\overline{W_3} + C_5^{\mathsf{in}}\overline{W_6} - C_6^{\mathsf{in}}\overline{W_5} + C_7^{\mathsf{in}}\overline{W_8} - C_8^{\mathsf{in}}\overline{W_7} + B_2\right)$$

$$C_3^{\mathsf{out}} = \sigma\left(C_1^{\mathsf{in}}W_3 - C_2^{\mathsf{in}}\overline{W_4} + C_3^{\mathsf{in}}W_1 + C_4^{\mathsf{in}}W_2 + C_5^{\mathsf{in}}\overline{W_7} - C_6^{\mathsf{in}}\overline{W_8} - C_7^{\mathsf{in}}\overline{W_5} + C_8^{\mathsf{in}}\overline{W_6} + B_3\right)$$

$$C_4^{\mathsf{out}} = \sigma\left(C_1^{\mathsf{in}}W_4 + C_2^{\mathsf{in}}W_3 - C_3^{\mathsf{in}}W_2 + C_4^{\mathsf{in}}W_1 + C_5^{\mathsf{in}}\overline{W_8} + C_6^{\mathsf{in}}\overline{W_7} - C_7^{\mathsf{in}}\overline{W_6} - C_8^{\mathsf{in}}\overline{W_5} + B_4\right)$$

$$C_5^{\mathsf{out}} = \sigma\left(C_1^{\mathsf{in}}W_5 - C_2^{\mathsf{in}}\overline{W_6} - C_3^{\mathsf{in}}\overline{W_7} - C_4^{\mathsf{in}}\overline{W_8} + C_5^{\mathsf{in}}W_1 + C_6^{\mathsf{in}}W_2 + C_7^{\mathsf{in}}W_3 + C_8^{\mathsf{in}}W_4 + B_5\right)$$

$$C_6^{\mathsf{out}} = \sigma\left(C_1^{\mathsf{in}}W_6 + C_2^{\mathsf{in}}W_5 - C_3^{\mathsf{in}}\overline{W_8} + C_4^{\mathsf{in}}\overline{W_7} - C_5^{\mathsf{in}}W_2 + C_6^{\mathsf{in}}W_1 - C_7^{\mathsf{in}}W_4 + C_8^{\mathsf{in}}W_3 + B_6\right)$$

$$C_7^{\mathsf{out}} = \sigma\left(C_1^{\mathsf{in}}W_7 + C_2^{\mathsf{in}}W_8 + C_3^{\mathsf{in}}W_5 - C_4^{\mathsf{in}}W_6 - C_5^{\mathsf{in}}W_3 + C_6^{\mathsf{in}}W_4 + C_7^{\mathsf{in}}W_1 - C_8^{\mathsf{in}}W_2 + B_7\right)$$

$$C_8^{\mathsf{out}} = \sigma\left(C_1^{\mathsf{in}}W_8 - C_2^{\mathsf{in}}W_7 + C_3^{\mathsf{in}}W_6 + C_4^{\mathsf{in}}W_5 - C_5^{\mathsf{in}}W_4 - C_6^{\mathsf{in}}W_3 + C_7^{\mathsf{in}}W_2 + C_8^{\mathsf{in}}W_1 + B_8\right)$$

Here, $W_i \in \mathbb{C}^{E \times E}$, $i = 1, \ldots, 8$ denotes the layer weights, $B \in \mathbb{C}^E$ represents the bias term, and the multiplication involves complex MLP operations across the embedding dimension.

# D ADDITIONAL ABLATION STUDIES

This section presents additional ablation studies, expanding on the findings reported in Section 5.3. We analyze the impact of FFT resolution, embedding size, and the number of STFT windows on WM-MLP performance. Additionally, we include further results for the frequency compression, sequence length, and real vs. imaginary component discussions. Furthermore, we provide a comparative analysis of various hyper-complex fields (octonions, quaternions, and sedenions) for the HC-MLP and report the corresponding results.

## D.1 PARAMETER SENSITIVITY

In this section, we conduct a parameter sweep to examine the effects of different hyperparameters on model performance. To accomplish this, we utilize two datasets: the ETTh1 dataset and the electricity dataset. Each section presents four graphs illustrating the results on the two datasets for a configuration of $I/O = 96 \times 96, 336$. Except for the specific experiment sweep, the embedding size is set to 128 for the ETTh1 dataset and 64 for the electricity dataset, with $M$ set to 0 for all datasets.

**Embed Size** In this section, we evaluate the influence of embedding size on the model's performance. We conducted experiments with embedding dimensions $E \in \{1, 2, 4, 8, 16, 32, 64, 128, 256, 512\}$, while keeping the following parameters fixed: $N_{\text{FFT}} = 16$, $B = 8$, $p = 13$, and $M = M_{\text{max}}$. We can observe that as we increase the embedding size, the loss decreases until we reach a certain point (which is dependent on the dataset). This is likely because a larger embedding size enables the model to capture more features; however, an excessively high embedding size may lead to overfitting.

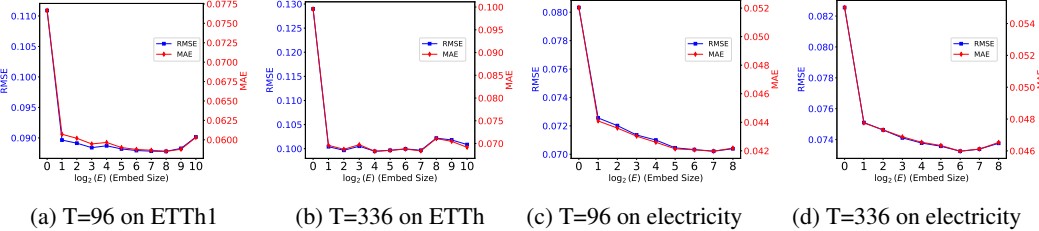

(a) T=96 on ETTh1  (b) T=336 on ETTh  (c) T=96 on electricity  (d) T=336 on electricity

Figure 7: Comparison of MSE and MAE across different values of E for varying T on the ETTh1 and Electricity datasets.

**Amount of Windows (High Dim)** In this section, we evaluate the influence of the number of windows ($p$) on the model's performance. We conducted experiments with different window counts $p \in \{3, 6, 14, 17, 25, 33\}$, while keeping the following parameters fixed: $B = 8$, $M = M_{\text{max}}$, and the overlap between windows is 50%.

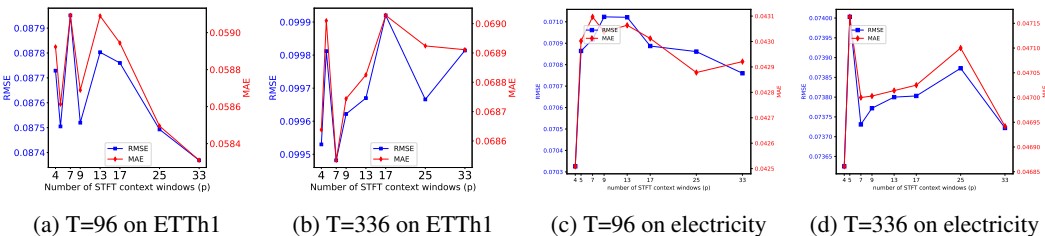

(a) T=96 on ETTh1  (b) T=336 on ETTh1  (c) T=96 on electricity  (d) T=336 on electricity

Figure 8: Comparison of MSE and MAE across different values of $p$ for varying $T$ on the ETTh1 and Electricity datasets.

**FFT Resolution (NFFT)** In this section, we evaluate the influence of the FFT resolution ($N_{\text{FFT}}$) on the model's performance. We conducted experiments with different $N_{\text{FFT}} \in$

$\{6, 8, 12, 16, 24, 32, 48\}$, while keeping the following parameters fixed: $p = 25$, $B = 8$, $M = M_{\max}$.

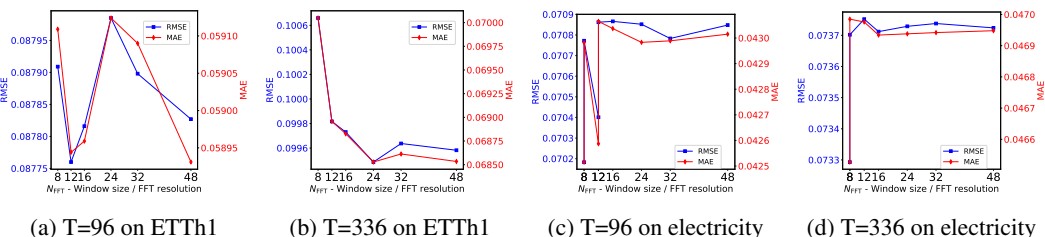

(a) T=96 on ETTh1 (b) T=336 on ETTh1 (c) T=96 on electricity (d) T=336 on electricity

Figure 9: Comparison of MSE and MAE across different values of $N_{FFT}$ for varying $T$ on the ETTh1 and Electricity datasets.

**Frequency Choose Max (M)** In this section, we provide additional results for various datasets and prediction lengths $T$ regarding the discussion on frequency compression 5.

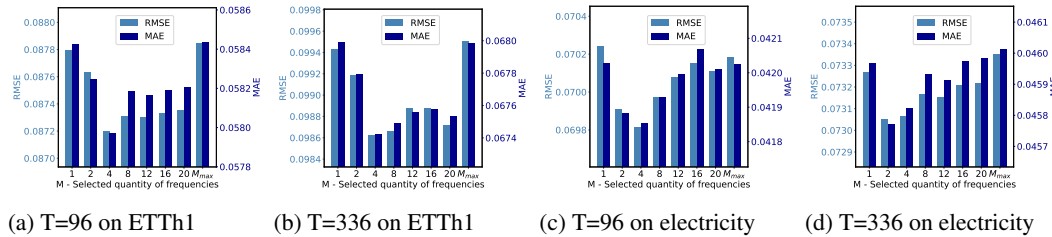

(a) T=96 on ETTh1 (b) T=336 on ETTh1 (c) T=96 on electricity (d) T=336 on electricity

Figure 10: Comparison of MSE and MAE across different values of M for various T on the ETTh1 and Electricity datasets.

## D.2 DIFFERENT LOOKBACK WINDOW

In this section, we present additional results for various lookback windows on the ETTh1 and ETTm1 datasets.

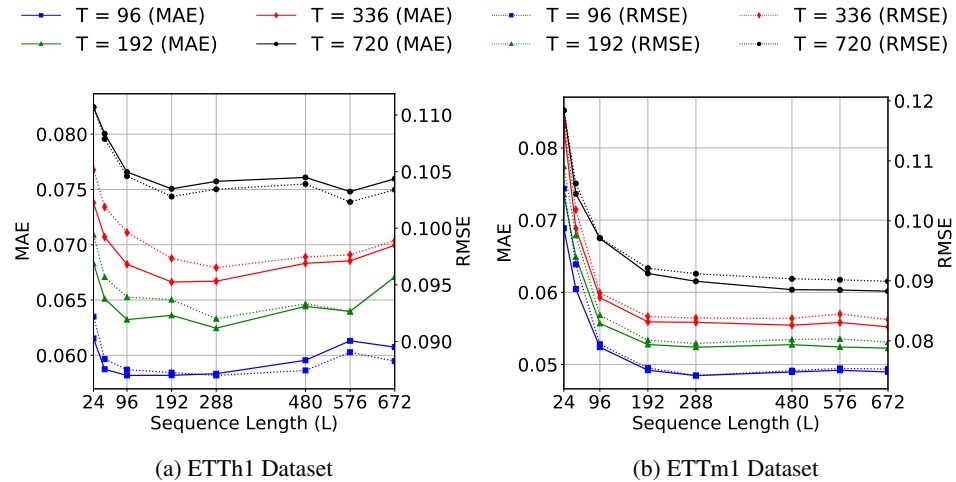

(a) ETTh1 Dataset (b) ETTm1 Dataset

Figure 11: MAE and RMSE in relation to the Lookback Window $L$ for varying prediction lengths $T \in \{96, 192, 336, 720\}$ for the ETTh1 and ETTm1 datasets.

## D.3 REAL VS IMAGINARY COMPONENTS

This section provides additional information regarding the real versus imaginary experiment discussed in Section 5.3.3.

| Dataset | I/O | 96/96 | | 96/192 | | 96/336 | | 96/720 | |
|---------|-----|-------|-----|--------|-----|--------|-----|--------|-----|
| | Hidden Part | MAE | RMSE | MAE | RMSE | MAE | RMSE | MAE | RMSE |
| ETTm1 | $X^{\text{Real}}$ | 0.0522 | 0.0797 | 0.0560 | 0.0850 | 0.0597 | 0.0888 | 0.0658 | 0.0958 |
| | $X^{\text{Imag}}$ | 0.0521 | 0.0792 | 0.0562 | 0.0844 | 0.0592 | 0.0879 | 0.0684 | 0.0976 |
| | $W^{\text{Real}}$ | 0.0522 | 0.0791 | 0.0557 | 0.0843 | 0.0588 | 0.0875 | 0.0669 | 0.0964 |
| | $W^{\text{Imag}}$ | 0.0526 | 0.0801 | 0.0560 | 0.0849 | 0.0596 | 0.0888 | 0.0651 | 0.0953 |
| | $W^{\text{Imag}}, X^{\text{Imag}}$ | 0.0523 | 0.0798 | 0.0560 | 0.0849 | 0.0592 | 0.0884 | 0.0644 | 0.0947 |
| | $W^{\text{Real}}, X^{\text{Real}}$ | 0.0522 | 0.0791 | 0.0557 | 0.0843 | 0.0588 | 0.0887 | 0.0669 | 0.0930 |
| | Normal | 0.0522 | 0.0791 | 0.0565 | 0.0848 | 0.0592 | 0.0878 | 0.0685 | 0.0975 |
| ETTh1 | $X^{\text{Real}}$ | 0.0584 | 0.0877 | 0.0638 | 0.0944 | 0.0684 | 0.0997 | 0.0767 | 0.1047 |
| | $X^{\text{Imag}}$ | 0.0582 | 0.0879 | 0.0634 | 0.0943 | 0.0679 | 0.0997 | 0.0756 | 0.1041 |
| | $W^{\text{Real}}$ | 0.0586 | 0.0880 | 0.0644 | 0.0948 | 0.0685 | 0.0998 | 0.0759 | 0.1039 |
| | $W^{\text{Imag}}$ | 0.0584 | 0.0880 | 0.0646 | 0.0951 | 0.0694 | 0.1008 | 0.0781 | 0.1065 |
| | $W^{\text{Imag}}, X^{\text{Imag}}$ | 0.0586 | 0.0880 | 0.0644 | 0.0947 | 0.0685 | 0.0998 | 0.0759 | 0.1040 |
| | $W^{\text{Real}}, X^{\text{Real}}$ | 0.0587 | 0.0882 | 0.0642 | 0.0948 | 0.0690 | 0.1005 | 0.0765 | 0.1050 |
| | Normal | 0.0586 | 0.0878 | 0.0639 | 0.0945 | 0.0684 | 0.0998 | 0.0765 | 0.1043 |

Table 8: Performance comparison on the ETTm1, ETTh1, and Electricity datasets for $I/O = 96 \times \{96, 192, 336, 720\}$ with different modes. $X^{\text{Real}}$ and $X^{\text{Imag}}$ refer to hiding the real and imaginary parts of the input, respectively. $W^{\text{Real}}$ and $W^{\text{Imag}}$ denote zeroing the real and imaginary weights, respectively. The cases where both the real and imaginary components are completely ignored (i.e., both weights and inputs are zeroed) are represented by $W^{\text{Imag}}, X^{\text{Imag}}$ and $W^{\text{Real}}, X^{\text{Real}}$. MAE and RMSE are reported, where lower values indicate better performance.

## D.4 HC-MLP EXPERIMENTAL RESULTS WITH FOR VARIOUS VALUES OF $p$

In this section, we present additional results on the HC-MLP for various bases. Specifically, we provide results for the Quaternion base ($p = 2$, QuatMLP), Octonion base ($p = 4$, OctMLP), and Sedenion base ($p = 8$, SedMLP). Additionally, we include results for a model that aggregates all windows without using hyper-complex numbers, referred to as BasicMLP. Further details about its implementation can be found in B.3.

| | Metric | Traffic | | | | ETTh1 | | | | ETTm1 | | | |
|---|--------|---------|-----|-----|-----|-------|-----|-----|-----|-------|-----|-----|-----|
| | | 96 | 192 | 336 | 720 | 96 | 192 | 336 | 720 | 96 | 192 | 336 | 720 |
| SedenionMLP ($p = 8$) | RMSE | 0.0340 | 0.0346 | 0.0351 | 0.0363 | 0.0896 | 0.0948 | 0.0999 | 0.1047 | 0.0814 | 0.0857 | 0.0894 | 0.0977 |
| | MAE | 0.0168 | 0.0169 | 0.0173 | 0.0186 | 0.0598 | 0.0640 | 0.0685 | 0.0767 | 0.0542 | 0.0573 | 0.0609 | 0.0682 |
| OctontionMLP ($p = 4$) | RMSE | 0.0335 | 0.0343 | 0.0349 | 0.0361 | 0.0834 | 0.0874 | 0.0941 | 0.1017 | 0.0739 | 0.0831 | 0.0888 | 0.0967 |
| | MAE | 0.0166 | 0.0167 | 0.0172 | 0.0185 | 0.0579 | 0.0635 | 0.0676 | 0.0759 | 0.0496 | 0.0556 | 0.0603 | 0.0673 |
| QuaternionMLP ($p = 2$) | RMSE | 0.0335 | 0.0343 | 0.0350 | 0.0362 | 0.0874 | 0.0938 | 0.0997 | 0.1059 | 0.0796 | 0.0847 | 0.0887 | 0.0974 |
| | MAE | 0.0165 | 0.0167 | 0.0172 | 0.0184 | 0.0580 | 0.0633 | 0.0687 | 0.0783 | 0.0526 | 0.0564 | 0.0603 | 0.0678 |
| BasicMLP | RMSE | 0.0372 | 0.0391 | 0.0384 | 0.0415 | 0.0962 | 0.1025 | 0.1061 | 0.1187 | 0.0832 | 0.0903 | 0.0967 | 0.1066 |
| | MAE | 0.0180 | 0.0195 | 0.0201 | 0.0217 | 0.0650 | 0.0714 | 0.0761 | 0.0886 | 0.0546 | 0.0595 | 0.0649 | 0.0753 |

Table 9: Comparison of different hypercomplex structures on the ETT and Traffic datasets. QuadMLP (2 windows), OctMLP (4 windows), and SedMLP (8 windows) represent hypercomplex models of increasing dimensionality, while BasicMLP is a non-hypercomplex linear model aggregating window information. Performance is reported using MSE and RMSE metrics, where lower values indicate better accuracy.

## D.5 EXTENDED NEIGHBORHOOD AGGREGATION IN WM-MLP

In this section, we present additional results on the WM-MLP with extended neighborhood aggregation. Specifically, we provide results for varying neighborhood sizes, where the model incorporates information not only from directly adjacent windows but also from second-order and third-order neighbors. The experiments were conducted on the ETTm1 and ETTh1 datasets with prediction lengths of 96, 192, 336, and 720.

For the two-neighbor case, the output $C_i^{\text{out}}$ is computed as:

$$
\begin{aligned}
C_i^{\text{out}} = \sigma\Big( & C_i^{\text{in}} W_{i \to i} + C_{i-1}^{\text{in}} W_{(i-1) \to i} + C_{i+1}^{\text{in}} W_{(i+1) \to i} \\
& + C_{i-2}^{\text{in}} W_{(i-2) \to i} + C_{i+2}^{\text{in}} W_{(i+2) \to i} + B_i \Big).
\end{aligned} \tag{7}
$$

For the three-neighbor case, the output $C_i^{\text{out}}$ is computed as:

$$
\begin{aligned}
C_i^{\text{out}} = \sigma\Big( & C_i^{\text{in}} W_{i \to i} + C_{i-1}^{\text{in}} W_{(i-1) \to i} + C_{i+1}^{\text{in}} W_{(i+1) \to i} \\
& + C_{i-2}^{\text{in}} W_{(i-2) \to i} + C_{i+2}^{\text{in}} W_{(i+2) \to i} \\
& + C_{i-3}^{\text{in}} W_{(i-3) \to i} + C_{i+3}^{\text{in}} W_{(i+3) \to i} + B_i \Big).
\end{aligned} \tag{8}
$$

| | | ETTh1 | | | | ETTm1 | | | |
|---|---|---|---|---|---|---|---|---|---|
| | **Metric** | 96 | 192 | 336 | 720 | 96 | 192 | 336 | 720 |
| WM-MLP (1 Neighbor) | RMSE | 0.084 | 0.088 | 0.097 | 0.102 | 0.076 | 0.082 | 0.089 | 0.094 |
| | MAE | 0.057 | 0.064 | 0.068 | 0.075 | 0.052 | 0.055 | 0.058 | 0.064 |
| WM-MLP (2 Neighbors) | RMSE | 0.095 | 0.101 | 0.106 | 0.120 | 0.084 | 0.091 | 0.097 | 0.104 |
| | MAE | 0.065 | 0.071 | 0.076 | 0.090 | 0.055 | 0.060 | 0.065 | 0.073 |
| WM-MLP (3 Neighbors) | RMSE | 0.095 | 0.102 | 0.109 | 0.120 | 0.084 | 0.091 | 0.097 | 0.010 |
| | MAE | 0.065 | 0.071 | 0.078 | 0.090 | 0.055 | 0.060 | 0.065 | 0.073 |

Table 10: Performance comparison of WM-MLP with varying numbers of neighbors (1, 2, and 3) on the ETTh1 and ETTm1 datasets for prediction lengths of 96, 192, 336, and 720. Metrics include RMSE and MAE. Results for WM-MLP with one neighbor are derived from the baseline values reported in the original paper.

## D.6 COMPLEXITY ANALYSIS

We conducted an asymptotic analysis of modern models to compare their training time, memory usage, and testing steps. The results are summarized in Table 11. The comparison highlights the computational efficiency of the WM-MLP and HC-MLP models relative to other state-of-the-art approaches. Specifically, both models demonstrate competitive performance with logarithmic complexity in training time and memory, and a constant number of testing steps.

| Method | Training Time | Training Memory | Testing Steps |
|---|---|---|---|
| WM-MLP | $\mathcal{O}(L \log \frac{L}{p})$ | $\mathcal{O}(L)$ | 1 |
| HC-MLP | $\mathcal{O}(L \log \frac{L}{P} + p^2)$ | $\frac{1}{3}\mathcal{O}(L)$ | 1 |
| FreTS | $\mathcal{O}(L \log L)$ | $\mathcal{O}(L)$ | 1 |
| PatchTST | $\mathcal{O}(L/S)$ | $\mathcal{O}(L/S)$ | 1 |
| LTSF-Linear | $\mathcal{O}(L)$ | $\mathcal{O}(L)$ | 1 |
| FEDformer | $\mathcal{O}(L)$ | $\mathcal{O}(L)$ | 1 |
| Autoformer | $\mathcal{O}(L \log L)$ | $\mathcal{O}(L \log L)$ | 1 |
| Informer | $\mathcal{O}(L \log L)$ | $\mathcal{O}(L \log L)$ | 1 |
| Transformer | $\mathcal{O}(L^2)$ | $\mathcal{O}(L^2)$ | L |
| Reformer | $\mathcal{O}(L \log L)$ | $\mathcal{O}(L \log L)$ | 1 |

Table 11: Comparison of models in terms of asymptotic complexity for training time, memory usage, and testing steps as a function of the lookback window length ($L$). Here, $S$ denotes the patch size used in PatchTST, and $p$ represents the number of windows in the STFT transformation.

## D.7 VISUALIZATIONS

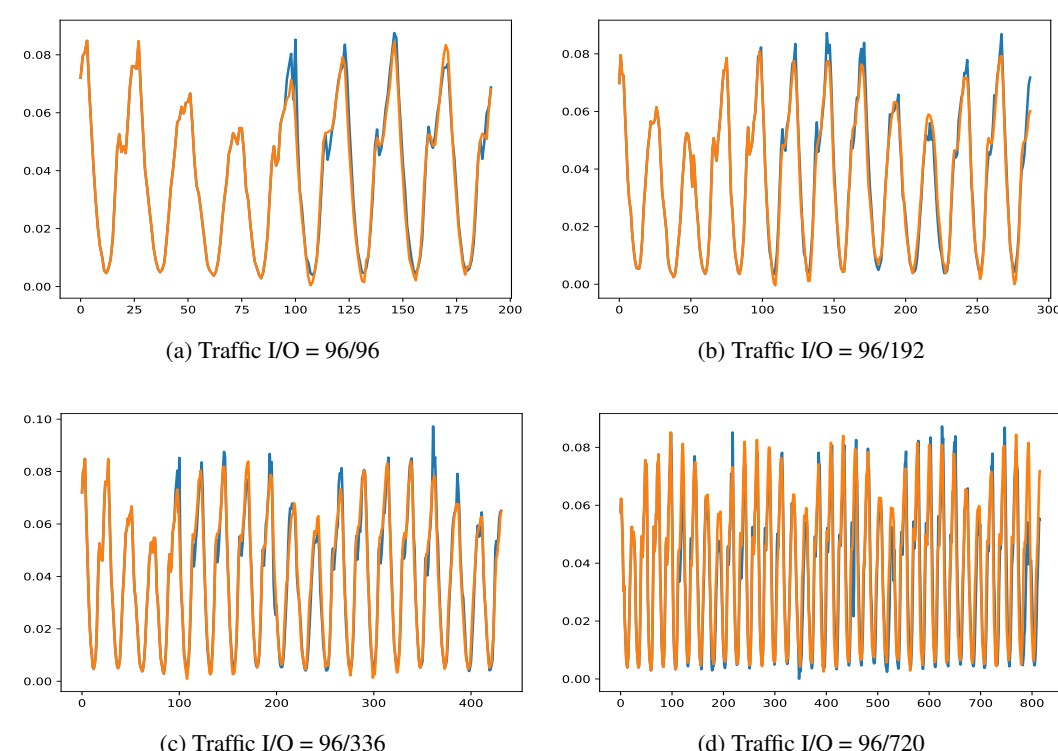

(a) Traffic I/O = 96/96

(b) Traffic I/O = 96/192

(c) Traffic I/O = 96/336

(d) Traffic I/O = 96/720

Figure 12: Ground Truth vs. Predictions for Different I/O Settings (Traffic Dataset).

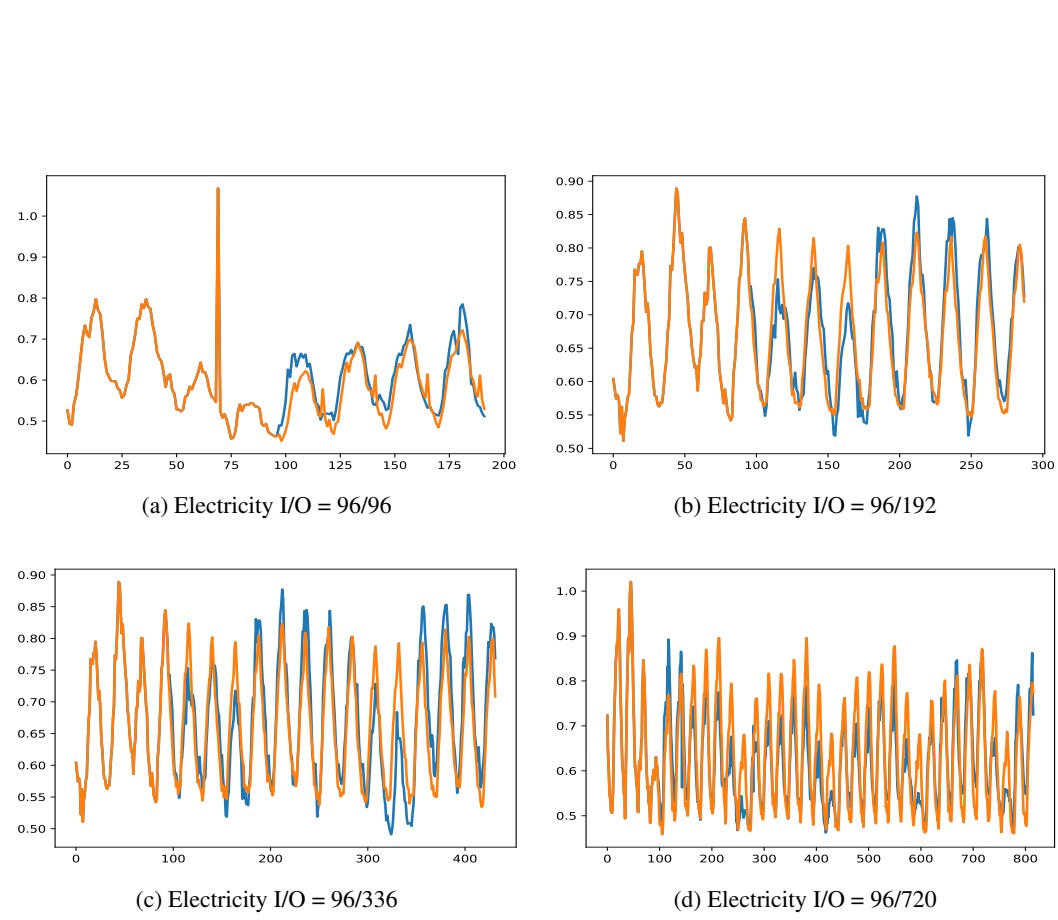

Figure 13: Ground Truth vs. Predictions for Different I/O Settings (Electricity Dataset).