# APPENDIX FOR "EFFICIENT TIME SERIES FORECASTING VIA HYPER-COMPLEX MODELS AND FREQUENCY AGGREGATION"

## A  NOTATIONS & SYMBOLS

### A.1  NOTATION

We provide a detailed table of the involved notation in this paper:

| Symbol | Description |
|--------|-------------|
| $B$ | Batch size. |
| $L$ | Lookback window size. |
| $D$ | Number of features for each time step. |
| $T$ | Length of the prediction horizon. |
| $E$ | Embedding size. |
| $M$ | Number of frequencies to select from all the frequencies using the top M magnitudes. |
| $X$ | Multivariate time series with a lookback window of size $L$ at timestamps $t$, where $X \in \mathbb{R}^{L \times D}$. |
| $X_t$ | Multivariate values of $D$ distinct series at timestamp $t$, where $X_t \in \mathbb{R}^D$. |
| $X_{t,i}$ | The value of the $i$-th feature of the distinct series at timestamp $t$, where $X_{t,i} \in \mathbb{R}$. |
| $\hat{X}$ | Ground truth target values, where $\hat{X} \in \mathbb{R}^{T \times D}$ |

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

 | 0.0165 | 0.0167 | 0.0172 | 0.0184 | 0.0580 | 0.0633 | 0.0687 | 0.0783 | 0.0526 | 0.0564 | 0.0603 | 0.0678 |

Table 9: Comparison between the WM-MLP and the HC-MLP of RMSE and MAE on ETTh1, Traffic, and ETTm1 datasets

## E.1 VISUALIZATIONS

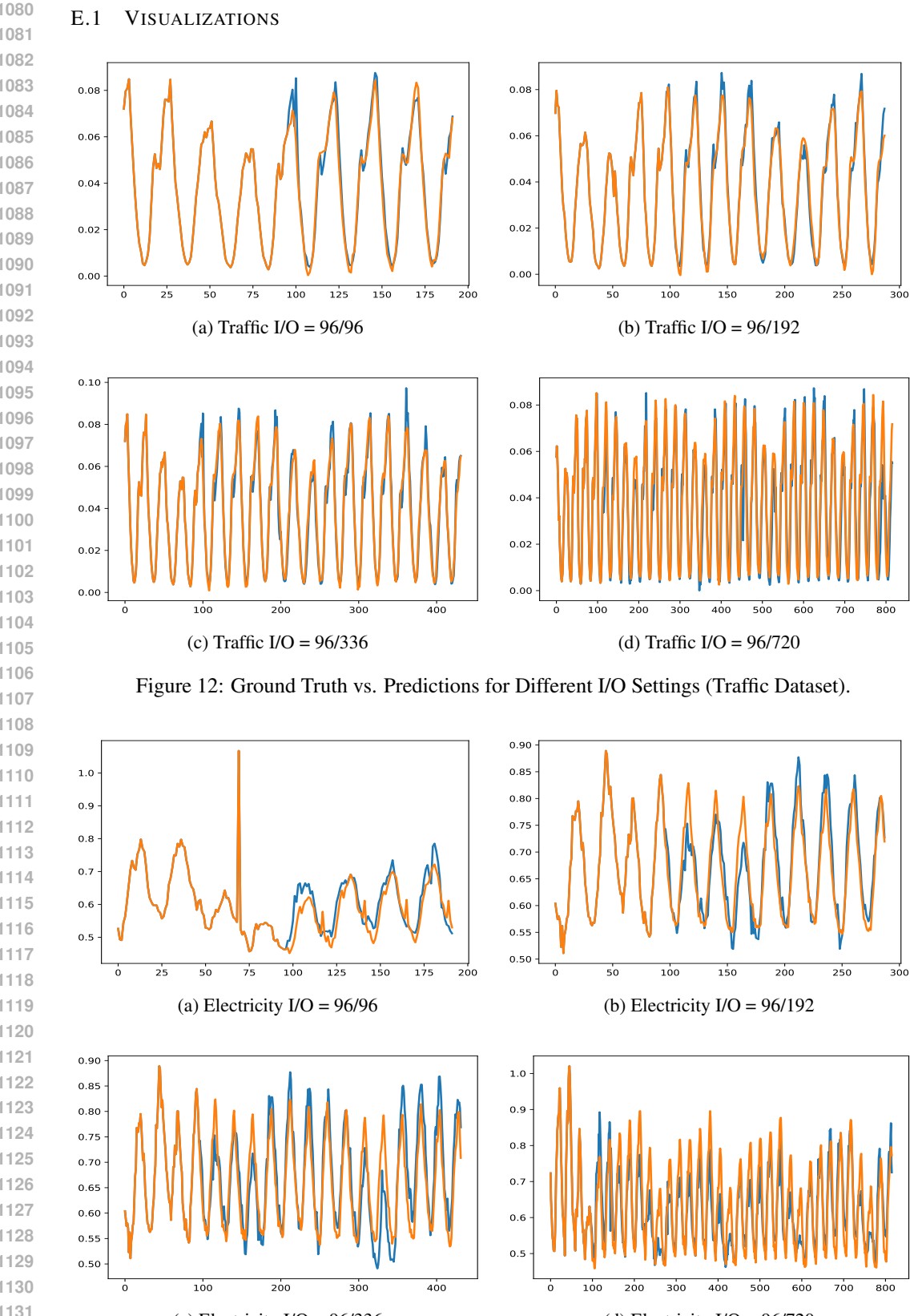

(a) Traffic I/O = 96/96

(b) Traffic I/O = 96/192

(c) Traffic I/O = 96/336

(d) Traffic I/O = 96/720

Figure 12: Ground Truth vs. Predictions for Different I/O Settings (Traffic Dataset).

(a) Electricity I/O = 96/96

(b) Electricity I/O = 96/192

(c) Electricity I/O = 96/336

(d) Electricity I/O = 96/720

Figure 13: Ground Truth vs. Predictions for Different I/O Settings (Electricity Dataset).