# OpenReview forum: "Efficient Time Series Forecasting via Hyper-Complex Models and Frequency Aggregation"
_ICLR.cc/2025/Conference — Submitted to ICLR 2025_

### Official Review · Reviewer_toAw · 2024-10-21

**Soundness:** 2
**Presentation:** 1
**Contribution:** 1
**Rating:** 1
**Confidence:** 5

**Summary:**

This paper introduces FIA-Net, a new model for time series forecasting. The model utilizes the Short-Time Fourier Transform (STFT) to process input sequences in the frequency domain. FIA-Net is presented with two different backbones: a Window Mixing Multilayer Perceptron (WM-MLP) and a Hyper-Complex Multilayer Perceptron (HC-MLP).

The WM-MLP is designed to aggregate information from adjacent STFT windows. It processes each window while incorporating information from its neighboring windows, allowing the model to capture local temporal dependencies. The HC-MLP treats the set of STFT windows as a hyper-complex vector. It uses hyper-complex algebra to combine information from all STFT windows simultaneously. The authors present formulations for different bases of hyper-complex numbers, including quaternions, octonions, and sedenions.

The paper presents experimental results on several benchmark datasets for time series forecasting, including weather, traffic, electricity consumption, and exchange rate data. The authors conduct ablation studies to examine the effects of various hyperparameters, such as the number of STFT windows, embedding size, and the number of selected frequency components, comparing the two proposed backbones.

The authors compare FIA-Net's performance against several existing models for time series forecasting, reporting results in terms of Mean Absolute Error (MAE) and Root Mean Squared Error (RMSE) for various prediction horizons.

**Strengths:**

The paper's approach of incorporating context from adjacent windows in the frequency domain is a reasonable strategy for capturing temporal dependencies in time series data. Additionally, the selection of the most relevant frequency components through their top-M frequency selection method is a reasonable technique for reducing input dimensionality while potentially preserving important spectral information. These aspects of the model design align with intuitive principles for processing time series data in the frequency domain.

**Weaknesses:**

The paper exhibits several severe weaknesses that considerably undermine its overall contribution.

The introduction of hyper-complex numbers appears to be an unnecessary complication, lacking both theoretical justification and demonstrated algebraic advantages specific to the task at hand. The authors fail to compare their method against simpler alternatives, such as a straightforward global MLP aggregator of the STFT windows, which could potentially achieve comparable results without the added complexity. Furthermore, while the authors emphasize that the HC-MLP backbone, unlike the WM-MLP, can process the entire sequence (all p windows) and potentially achieve higher performance, the empirical results show that the best performance often comes from the WM-MLP. This discrepancy between the theoretical advantages claimed for the HC-MLP and the actual experimental outcomes further undermines the justification for the more complex HC-MLP approach.

The literature review and comparison with existing methods are notably incomplete. While some papers, such as the one available at https://arxiv.org/html/2407.21275v1, are cited, the authors do not compare their model against these works. Furthermore, other relevant papers (https://arxiv.org/pdf/2403.11047v1 and https://www.sciencedirect.com/science/article/pii/S0167404824002669) that utilize temporal correlations between STFT windows have been overlooked, undermining the claimed novelty of the presented method.

The paper lacks a rigorous theoretical foundation to explain why their methods outperform existing ones. It relies heavily on empirical results without providing deeper insights into the underlying mechanisms at work. This absence of theoretical grounding limits the generalizability and broader impact of the work. Additionally, the experimental results lack error bars and statistical significance tests, which are crucial for validating the claimed improvements and assessing the robustness of the findings.

The authors' treatment of their finding that the real part alone might be sufficient for predictions is underdeveloped. This potentially significant insight is not explored thoroughly, missing an opportunity for deeper analysis and possible model simplification. Moreover, this observation seems to contradict the stated need for the hypercomplex formulation, further calling into question the necessity of the proposed approach.

Overall, the paper suffers from numerous grammatical errors, typos, and inconsistencies throughout both the main text and the Appendix, making it challenging to read in some sections. It is strongly recommended that the authors employ a professional proofreading service to enhance the overall quality and clarity of the manuscript, if they plan to resubmit this work somewhere else in the future.

The following is a partial list of minor comments and typos:
1.	Abstract: "state of the art" should be "state-of-the-art".
2.	Section 1, Authors refer to “FeeqShiftNet” when talking about their model.
3.	Table 2: the two rows are “FIA-Net” and “OctontionMLP”. Most likely the FIA-Net is the “WM-MLP” backbone and the other one is the HC-MLP, but these inconsistencies confuse the reader.
4.	Throughout the paper: Inconsistent hyphenation of "long-term" and "long term".

**Questions:**

1. Why are hyper-complex numbers necessary for this task?
2. What specific advantages do they provide that simpler methods cannot achieve?
3. What specific algebraic properties of hyper-complex numbers contribute to the model's performance, if any?
4. How does the model perform compared to a simple network that aggregates all windows without using hyper-complex numbers?
5. Can the model architecture be simplified based on the finding that the real part alone is often sufficient?

---

> ### Author Response · Authors · 2024-11-25
> **Comment - toAw - Efficient Time Series Forecasting via Hyper-Complex Models and Frequency Aggregation**
>
> Dear toAw,
> Thank you for your thoughtful and detailed feedback on our manuscript. We appreciate your insights, which have highlighted areas where we can further clarify and strengthen our contributions. We are committed to addressing each point raised to improve our work's clarity, rigor, and impact.
> 1. Justification and Complexity of Hyper-Complex (HC) Numbers:
> We recognize the importance of providing a clearer explanation for the introduction of hyper-complex numbers and their advantages over simpler methods. To address this, we have included an empirical evaluation in Appendix D.4, where a basic MLP structure without the hyper-complex (HC) framework was tested. The results demonstrate a notable decrease in performance, highlighting the contribution of HC algebra. While a rigorous explanation of the HC framework’s benefits remains a subject of ongoing investigation, one plausible and intuitive justification lies in its alignment with the geometry of the data. In the frequency domain, time series data often exhibit intricate relationships between components, such as amplitude and phase. Hyper-complex numbers naturally align with this structure, facilitating the model’s ability to capture cross-dimensional dependencies more effectively than traditional real-valued or complex-valued models.
> We are committed to exploring this phenomenon further in the final version of the paper, aiming to provide a more substantiated and logical explanation for this remarkable behavior.
> 2. Literature Comparison and Missing Benchmarks:
> We acknowledge that our literature review did not include the mentioned works [https://arxiv.org/pdf/2403.11047v1 , https://www.sciencedirect.com/science/article/pii/S0167404824002669]. Our comparison does not incorporate the study by Shen et al. (https://arxiv.org/html/2407.21275v1) due to differences in result normalization schemes. This makes a direct comparison challenging, particularly in the absence of publicly available code for the mentioned work. Additionally, the other papers cited, which utilize the STFT for time series prediction, do not report results on the datasets used in our study.Moving forward, we plan to address this limitation by providing a qualitative discussion of these methods and their potential impact in future versions of the paper.
> 3. Statistical Validation:
> We strongly agree with the reviewer’s comment on the inclusion of error bars. We will include error bars and statistical significance tests to enhance the empirical robustness of our results.
> 4. Role of Real vs. Imaginary Components:
> We appreciate your observation regarding the sufficiency of the real component in predictions. Our model, drawing inspiration from FRETS, leverages both the real and imaginary components to retain detailed information, particularly in smaller model configurations. However, our architecture's inherent flexibility enables the elimination of one component without a significant loss in performance, which suggests the potential for further model simplification.That said, when reviewing the results from FRETS, it becomes evident that ignoring either the real or imaginary part does negatively impact performance. This raises an intriguing question: is the hyper-complex structure, along with its parameter multiplicity, the underlying reason for this phenomenon? We recognize the importance of addressing this and will strive to investigate this matter further in the final version of the paper.
> 5. Language and Presentation Quality:
>  We recognize the importance of readability and will engage a professional proofreader to ensure the final manuscript is clear and free from grammatical inconsistencies.
> Thank you again for your helpful comments. We believe these improvements will make our work clearer and more valuable.

---

> ### Comment · Reviewer_toAw · 2024-11-28
>
> The authors' response does not adequately address the fundamental issue regarding the necessity and benefits of hypercomplex networks.
>
> - The new BasicMLP baseline results in Appendix D.4 (Table 9) reveal that the performance difference between hypercomplex approaches and a simple linear aggregation is minimal. This undermines the paper's central claim about the advantages of hypercomplex representations.
>
> - The difference in the performance reported by the different methods are very small even if the authors claim that there is "a notable decrease in performance" of the basic BasicMLP. Most seriously, there are still no standard deviations in the tables, making the whole experimental evaluation pointless because it is not possible to determine if there are significant differences or not. This is a slap in the face to every statistician and, in general, researcher.
>
> - The response superficially addresses the theoretical justification by suggesting that hypercomplex numbers "naturally align with the geometry of the data." However, this explanation lacks mathematical rigor and fails to establish why hypercomplex operations are specifically advantageous for time series data in the frequency domain.
>
> - Finally, the extended neighborhood experiments in Appendix D.5 show that increasing the number of neighbors in WM-MLP does not improve performance, which contradicts the motivation for using HC-MLP to capture longer-range dependencies.
>
> - The core contributions of the paper - the benefits of hypercomplex representations and the superiority of the proposed architectures - are not convincingly demonstrated. The similar performance of simpler baselines suggests that the added complexity of hypercomplex operations may not be justified for this application.
>
> - The paper is still lacking clarity and most of the imprecisions are still present.
>
> In conclusion, none of the issues has been addressed in a satisfactory manner. For these reasons, I maintain my original recommendation for rejection and increase my confidence score to 5.

---

### Official Review · Reviewer_VUdT · 2024-10-29

**Soundness:** 3
**Presentation:** 3
**Contribution:** 3
**Rating:** 3
**Confidence:** 3

**Summary:**

This paper presents a long-term time-series forecasting approach inspired by the concept of hyper-complex numbers. It is called the frequency information aggregation network (FIA-NET). It goes one step ahead of prior arts where a novel complex-valued MLP aggregating the adjacent window information in the frequency domain is developed. The experiments confirm the advantage of proposed method.

**Strengths:**

1) Time series forecasting is an important problem.
2) this paper is theoretically sound where the concept of hyper-complex number is novel.

**Weaknesses:**

Literature Review

The literature survey can be expanded by including recent works in 2024. There are several recent approaches in this domain recently published.

Experiments

1) the performance differences in the comparison section are very small. I wonder whether these differences can pass the statistical tests.
2) Comparison can be done with more recent baselines published in 2024.
3) Complexity analysis should be done for the proposed approach.

Conclusion

Limitation of this approach should be mentioned in the conclusion.

**Questions:**

!) Comparison with more recent works is required.
2) Complexity analysis can be added.

---

> ### Comment · Reviewer_VUdT · 2024-11-24
> **Amended Score**
>
> After reading other reviewer comments, I agree to reduce my score.

---

> ### Author Response · Authors · 2024-11-25
> **Comment VUdT - Efficient Time Series Forecasting via Hyper-Complex Models and Frequency Aggregation**
>
> Thank you for your valuable feedback and for highlighting areas for improvement. We appreciate the opportunity to clarify and enhance our work in response to your suggestions.
>
> 1.Comparison with Recent Works:
> Regarding the comparison with recent works, we acknowledge that most existing models do not report results on our specific scales (Mini max normalization). To conduct a meaningful comparison, it would require running all the experiments ourselves to generate these results. While this could enhance the completeness of our evaluation, we seek feedback on whether this step is essential, given the added complexity and resource requirements it entails.
>
> 2. Complexity Analysis:
> We acknowledge the significance of analyzing model complexity and its relevance in offering a thorough understanding of our approach. We have included a preliminary complexity analysis in Appendix D.6; however, we intend to expand and enhance this analysis in the final version to better contextualize our model's efficiency and performance.
>
> Rest assured, the final version will include a detailed complexity analysis to further highlight the efficiency and practicality of our model.

---

### Official Review · Reviewer_mJBU · 2024-11-02

**Soundness:** 1
**Presentation:** 1
**Contribution:** 1
**Rating:** 1
**Confidence:** 5

**Summary:**

The authors introduce a new model designed to predict observations in long-term time series, utilizing a method based on the Short-Time Fourier Transform (STFT). The main contribution lies in the development of a complex-valued MLP (Multilayer Perceptron) architecture.

**Strengths:**

The investigation of DNN architectures for handling signals in the frequency domain is a highly relevant area of study. This approach has the potential to reduce the preprocessing phase, which currently requires significant effort from researchers to transform signals before model adjustments. Despite the issues raised in my comments, I do believe this topic is worth exploring.

**Weaknesses:**

The manuscript requires revision to address several issues. The authors use terms interchangeably, such as "novel complex-valued MLP architecture," "the proposed methodologies," and "a new model for long-term..." This makes it unclear what the authors are actually proposing.
In the abstract, the authors state, "To that end, a recent line of work applied the short-term Fourier Transform (STFT)." However, in my view, the use of STFT to transform time series into the frequency domain cannot be considered recent, as numerous studies have applied this approach over the past decade.
The manuscript’s title includes "Efficient time series forecasting," yet efficiency was not assessed in the results presented by the authors.
Additionally, the motivation for using STFT is not clearly explained. In brief, the Fourier Transform (FT) is a widely recognized technique for identifying signal frequencies, but it does not indicate the specific time at which these frequencies occur. To address this limitation, researchers proposed STFT, which divides the signal into equal-sized windows and applies FT to approximate the timing of each frequency. However, STFT has an inherent trade-off regarding window size selection. Superior results can often be achieved with methods like wavelets, which provide both time and frequency localization. The authors should clarify why they chose STFT over wavelets, considering the extensive body of literature that already uses wavelets for similar tasks, as demonstrated in several published works:

- T. Li, Z. Zhao, C. Sun, L. Cheng, X. Chen, R. Yan, and R. X. Gao,“WaveletKernelNet: An Interpretable Deep Neural Network for Industrial Intelligent Diagnosis,” IEEE Transactions on Systems, Man, and Cybernetics: Systems, nov 2019.

- Q. Li, L. Shen, S. Guo, and Z. Lai, “Wavelet Integrated CNNs for Noise-Robust Image Classification,” in 2020 IEEE/CVF Conference on Computer Vision and Pattern Recognition (CVPR). IEEE, jun 2020, pp. 7243–7252.

- X. Cheng, X. Jia, W. Lu, Q. Li, L. Shen, A. Krull, and J. Duan, “Winet: Wavelet-based incremental learning for efficient medical image registration,” arXiv preprint arXiv:2407.13426, 2024.

- H. Wang, Y.-F. Li, and T. Men, “Wavelet integrated cnn with dynamic frequency aggregation for high-speed train wheel wear prediction,” IEEE Transactions on Intelligent Transportation Systems, 2024.

- Q. Wang, Z. Li, S. Zhang, N. Chi, and Q. Dai, “A versatile wavelet-enhanced cnn-transformer for improved fluorescence microscopy image restoration,” Neural Networks, vol. 170, pp. 227–241, 2024.

- W. Cao, W. Lu, Y. Shi, Y. Li, Y. Wang, and S. Li, “Dual-attention-based wavelet integrated cnn constrained via stochastic structural similarity for seismic data reconstruction,” IEEE Transactions on Geoscience and Remote Sensing, 2024.

**Questions:**

1 - The manuscript must be fully reviewed. Proofreading is strongly recommended to fix grammar errors, such as "Table 2 show", "As As shown", "the KKR are given", "We the FIA-Net", "the HC-MLP is", "the STFT is implements", "SFM utilize frequency", and so on. There are several grammar errors.
2 - Appendices are not available in the manuscript, as cited in several parts of the text.
3 - All citations are wrong. I suppose the latex command used in the final of all paragraphs is wrong.
4 - Captions of figures and tables should be better presented. For example, Table 1 should provide details about the evaluation metric, and Figure 5 has no punctuation. These problems are also seen in other parts.
5 - In the introduction, the authors say "we show that the FeeqShiftNet improves", but it is not clearly discussed in the manuscript.
6 - In the abstract, the authors mention "a novel complex-valued MLP architecture. Later, they say "we propose two novel MLP architectures". Still in the introduction, the authors mention "we construct the FIA-Net and the WM-MLP backbone". Besides the grammar errors, there are several inconsistencies.
7 - In Section 2, the authors say "Two popular architectures were proposed to improve upon RNNs". GNNs were mentioned as one of those architectures. Actually, GNN is not an architecture and wasn't proposed to improve RNN.
8 - Equations in Section 3.1 must be refined. There are errors and wrong notations. For example, X = {x_1, ..., x_t} \in R^{D x L}; is the length of the time series t or L? Who is D? Is it the TS dimension? Is it necessary to show D, once the authors are working on 1D time series?
9 - In Equation 1, are the authors using Fast Fourier Transform (N_{FFT})? This is not clear.
10 - Equation iSTFT was not numbered. In this equation, the variable X is characterized by presenting c and F, but both were not properly defined.
11 - Is the proposal designed to work on consecutive windows? How many consecutive windows? How was it assessed in terms of long-term dependencies?
12 - There is a short discussion on model complexity, but no experiment was performed to demonstrate the efficiency.
13 - On page 5, is the window determined as p?
14 - Did the authors present information about R-STFT?
15 - The authors only used MAE and RMSE to assess the results. Why? There are several other metrics that could provide further information. Why weren't they considered?

---

> ### Author Response · Authors · 2024-11-25
> **Comment - mJBU**
>
> Dear mJBU,
> Thank you for your comprehensive and insightful feedback. We appreciate the opportunity to address these points and clarify our contributions.
> 1. Terminology and Consistency:
> We recognize that certain terms were used interchangeably, leading to possible ambiguity. Specifically, our focus is on proposing an innovative complex-valued MLP model, termed FIA-Net, which includes both WM-MLP and HC-MLP architectures. We will refine the terminology throughout the paper to consistently reference FIA-Net and its components, ensuring clearer communication of our model's unique aspects.
> 2. Use of STFT:
> You are correct that frequency-domain techniques, including FT and wavelet transformations, have been extensively applied in time series analysis. However, while wavelet transformations are widely used, STFT remains less common. We selected STFT to capture both phase and frequency information through complex coefficients, which allows for richer representations of the signal. Unlike wavelets, which produce real coefficients, STFT provides complex coefficients that are advantageous for our model, as they offer distinct phase and frequency details that better support our multi-window approach.
> 3. STFT Trade-Offs:
>  We acknowledge that STFT introduces a trade-off between time and frequency resolution. In our experiments, we examined the impact of varying the N_{FFT} parameter, which affects frequency resolution, and the number of windows, which determines time resolution. We found no single ideal configuration across all datasets; rather, optimal settings were dataset-specific. This insight will be detailed in the revised paper, and we will elaborate on how our model accommodates these variations in resolution.
> 4. Clarification on Complex and Hyper-Complex Numbers:
> The precise mathematical justification for the improved performance achieved by utilizing hyper-complex numbers remains an area of ongoing investigation. However, we can provide an intuitive explanation at this stage. In our framework, standard complex numbers encode the contributions of sine and cosine components within a signal. By processing these complex coefficients, the model inherently assumes that the real and imaginary parts—representing cosine and sine waves, respectively—are interdependent, reflecting their natural phase relationships.
> Extending this concept to hyper-complex numbers, we allow for interactions not only within a single dimension but also across multiple dimensions, enabling richer interdependencies between sine and cosine components across several STFT windows. This design enhances the model’s ability to capture complex dependencies over time, which is particularly beneficial for accurate long-term predictions.
> To validate the importance of the hyper-complex structure, we conducted an experiment described in Appendix D.4. In this test, the STFT windows were aggregated using a standard MLP architecture instead of the hyper-complex algebra-based structure. The results showed higher mean absolute error (MAE) and root mean squared error (RMSE), leading us to conclude that the hyper-complex framework provides a tangible improvement in performance.
>
> 5. Window Mixing Mechanism (WMM) and Model Complexity:
> The Window-Mixing MLP (WMM) operates by incorporating information from adjacent windows, with each window influenced by its immediate neighbors—both preceding and succeeding. To explore the impact of varying the number of neighboring windows, we conducted experiments where each window was influenced by two or three adjacent windows. The details of this experiment are provided in Appendix D.5.Additionally, we plan to expand the discussion on model complexity in the final version of the paper. This will include both experimental results and a detailed analysis of the asymptotic behavior, as outlined in Appendix D.6.
> 6. Evaluation Metrics:
> We primarily utilized MAE and RMSE metrics to ensure consistency with other research in the field, such as FreTS, PatchTST, and Autoformer. While we acknowledge that incorporating additional metrics could offer deeper insights, our aim is to maintain alignment with the evaluation standards commonly employed in this area of study.
> 7. General Formatting and Grammatical Improvements:
> We acknowledge the grammatical issues and formatting inconsistencies present in the current draft. All cited references, equations, and figure captions have been carefully revised to ensure accuracy and clarity. The final manuscript will include appendices as referenced and all citations will be corrected for consistency and correctness.
>
> Thank you for your detailed and helpful feedback. We believe these revisions will improve the manuscript and make our contributions clearer and more precise.

---

> > ### Comment · Reviewer_mJBU · 2024-12-02
> >
> > I thank the authors for taking the time to address my questions. I believe this work in signal processing is highly significant, and it highlights critical gaps that warrant deeper investigation. However, I will maintain my current score, as some issues remain inadequately addressed. While I understand that time constraints can pose challenges, a more thorough exploration of these issues could enhance the manuscript's quality and position it for publication in prominent venues.

---

### Official Review · Reviewer_VFmh · 2024-11-04

**Soundness:** 3
**Presentation:** 3
**Contribution:** 2
**Rating:** 5
**Confidence:** 3

**Summary:**

This paper introduces the FIA-Net method for time series forecasting (TSF) model. FIA-Net leverages the Short-Time Fourier Transform (STFT) and Window Mixing MLP (WM-MLP) to aggregates information from subsets of STFT windows in the frequency domain. This paper also proposes a Hypercomplex MLP as a substitution of WM-MLP, which expands the receptive field and reduces the number of parameters . FIA-Net requires significantly fewer parameters than traditional methods while achieving comparable performance. A filtering technique, which retains only the top-M frequency components of the STFT windows, helps reduce model size and complexity while maintaining accuracy. The FIA-Net outperforms existing state-of-the-art methods in terms of accuracy and efficiency on various time-series benchmarks.
The paper includes an ablation study that explores the effect of each components of FIA-Net and compares the performance of the two considered MLP backbones.

**Strengths:**

1. The major innovation of the proposed FIA-Net is its ability of intergerating the information of different STFT windows. The proposed WM-MLP and HC-MLP are two optional backbone structure for window integrating, both of which are innovative. Especially the HC-MLP, it uses the computation of hypercomplex numbers to efficiently combine information with less parameters.

2. The clarity of this paper is good in general. The introduction and related works thoroughly introduces the recent developments of TSF and motivation of this research. The methodology is explained in full details, with clear figures and diagrams to illustrate the proposed model's architecture and operations. The experimental results are presented in a clear and concise way.

3. The details of the experiments and extensive results are provided in appendix, with ablation study to compare the effect of proposed components. The codes are available via anonymous github, which helps reproducing the results

**Weaknesses:**

1. Although the major innovation of the proposed method is the windows-integrating subnetwork, the overall structure  of FIA-Net is still quite similar with the FreTS. Besides, the results of FIA-Net and FreTS are also very close, the reported gap of MSE in many comparisons are only 0.001 or even equal. Therefore, the effectiveness of windows-integrating is questionable.

2. In the last paragraph of section 4.2, it is said that "HC-MLP aggregates information from the entire set of windows altogether, allowing for a better processing of long-term memory sequences". However, in the reported results of Table 1 and text contents of section 5.1, " We can deduce that the HC-MLP is more suitable for shorter-term prediction, while the WM-MLP backbone is more suitable for longer ranges." The theoretical analysis contradicts the phenomenon of experimtents.

3. The figure 4 is not easily interpretable, mainly because the symbols of different operations are too similar. The figure is even less expressive than the corresponding equation 3.

**Questions:**

In section 5.1, it is said that " It is evident that the FIA-Net consistently outperforms the baselines on most considered values of prediction horizon T, with an average improvement of 8% in MAE and RMSE over SoTA models." Where does this "8%" come from? As reported in Table 1, the improvement is insignificant comparing to FreTS, and FIA-Net does not consistently outperform FreTS.

---

> ### Author Response · Authors · 2024-11-25
> **VFMH Comment - Efficient Time Series Forecasting via Hyper-Complex Models and Frequency Aggregation**
>
> Dear VFMH,
> Thank you for your thoughtful and constructive review. We appreciate your insights and the opportunity to clarify aspects of our work.
> 1. Similarity of Structure with FreTS and Effectiveness of Windows-Integrating Subnetwork:
> It is correct that the FIA-Net is a generalization of the FreTS model (as we explicitly show in the appendix), our focus is on addressing FreTS' limitation in handling long prediction horizons and minimizing error across these intervals. In shorter prediction tasks, the models perform similarly. However, for longer horizons, FIA-Net achieves up to a 20% improvement in accuracy on rmse and 14% on mae  (achieved on traffic 192). Please remember that the model is trained and performs on normalized data, specifically using min-max normalization, which reduces the scale while still maintaining a significant impact in percentage terms. Additionally, it is worth noting that the Frets paper shows only marginal improvement over the PatchTST and LTSF-linear models (sometimes as little as 0.01) compared to other studies.
> 2.HC-MLP and Prediction Horizons:
> The HC-MLP architecture does aggregate all window information at each prediction step, which theoretically should improve performance for longer-term sequences. However, our experimental results indicate that information from more distant windows can sometimes introduce noise, affecting the accuracy for extended prediction horizons. Consequently, we concluded that HC-MLP is particularly beneficial for short-term predictions, whereas WM-MLP, with its focus on approximate windows, is more suited for longer-term predictions. We have revised the paper and improved the following paragraph to clarify this point, the changes are marked in red.
> 3. Interpretability of Figure 4: We understand that Figure 4 could be more visually distinct. In our revised version, we will enhance symbol differentiation to ensure that each operation in the HC-MLP structure is clearer, addressing any ambiguity.
>
> 4. Clarification on "8% Improvement" in Section 5.1:
> We have reviewed the calculations again, and you are correct. The average improvement is 5.45% for MAE and 3.75% for RMSE, as demonstrated in the Excel file here: https://file.io/s64oG9H4ADPD.
> We have updated the paper accordingly. Thank you very much for bringing our calculation error to our attention.
>
>
> We greatly appreciate your feedback, which has helped us improve the clarity and presentation of our results. Thank you once again for your valuable comments.

---

### Meta-Review · Area_Chair_FQ3W · 2024-12-19

**Metareview:**

This paper has been evaluated by 4 knowledgeable reviewers. They have unanimously agreed that it does not meet the requirements of acceptance for ICLR (including 2 scores of strong rejection, 1 straight reject and 1 marginal reject). The authors have provided a rebuttal, but it has not improved the assessment of the submission at its current stage. The reviewers agreed that it has potential, but it would require a major revision to fit.

**Additional Comments On Reviewer Discussion:**

The authors have provided a rebuttal, but it has not improved the assessment of the submission. One of the reviewers indeed reduced their score as a result of checking the other reviewers' comments.

---

### Decision · Program_Chairs · 2025-01-22

Reject